# Aerosol optical properties over Europe: an evaluation of the AQMEII Phase 3 simulations against satellite observations

Laura Palacios-Peña[1], Pedro Jiménez-Guerrero[1], Rocío Baró[1a], Alessandra Balzarini[2], Roberto Bianconi[3], Gabriele Curci[4,5], Tony Christian Landi[6], Guido Pirovano[2], Marje Prank[7,8], Angelo Riccio[9], Paolo Tuccella[4,5], and Stefano Galmarini[10]

[1]Physics of the Earth, Department of Physics, Regional Campus of International Excellence (Campus Mare Nostrum), University of Murcia (UMU-MAR), 30100 Murcia, Spain
[2]Ricerca sul Sistema Energetico (RSE SpA), Milano, Italy
[3]Enviroware srl, Concorezzo, MB, Italy
[4]CETEMPS, University of L'Aquila, Italy
[5]Dept. Physical and Chemical Sciences, University of L'Aquila, Italy
[6]CNR - Institute for Atmospheric Sciences and Climate, Bologna, Italy
[7]Finnish Meteorological Institute, Atmospheric Composition Research Unit, Helsinki, Finland
[8]Cornell University, Atmospheric and Earth Sciences, Ithaca, NY, USA
[9]University Parthenope of Naples, Dept. of Science and Technology, Napoli, Italy
[10]European Commission, Joint Research Centre (JRC), Directorate for Energy, Transport and Climate, Air and Climate Unit, Ispra (VA), Italy
[a]Now at: Section Chemical Weather Forecasts, Division Data/Methods/Modelling, ZAMG – Zentralanstalt für Meteorologie und Geodynamik, Vienna, Austria

*Correspondence to:* Pedro Jiménez-Guerrero (pedro.jimenezguerrero@um.es)

**Abstract.** The main uncertainties regarding the estimation of changes in the Earth's energy budget are related to the role of atmospheric aerosols. These changes are caused by aerosol-radiation (ARI) and aerosol-cloud interactions (ACI), which heavily depend on aerosol properties. Since the 1980s, many international modelling initiatives have studied atmospheric aerosols and their climate effects. Phase 3 of the Air Quality Model Evaluation International Initiative (AQMEII) focuses on evaluating and

intercomparing regional and linked global/regional modelling systems by collaborating with the Task Force on the Hemispheric Transport of Air Pollution Phase 2 (HTAP2) initiative. Within this framework, the main aim of this work is the assessment of the representation of aerosol optical depth (AOD) and the Ångström exponent (AE) in AQMEII Phase 3 simulations over Europe. The evaluation was made using remote sensing data from the Moderate Resolution Imaging Spectroradiometer (MODIS) sensors onboard the Terra and Aqua platforms; and the instruments belonging to the ground-based networks Aerosol Robotic

Network (AERONET) and the Maritime Aerosol Network (MAN). Overall, the skills of AQMEII simulations when representing AOD (mean absolute errors from 0.05 to 0.30) produced lower errors than for the AE (mean absolute errors from 0.30 to 1). Regardless of the models or the emissions used, models were skillful at representing the low and mean AOD values observed (below 0.5). However, high values (around 1.0) were overpredicted for biomass burning episodes, due to an underestimation in the common fires' emissions; and were overestimated for coarse particles -principally desert dust- related to the boundary

conditions. Despite this behaviour, the spatial and temporal variability of AOD was better represented by all the models than AE variability, which was strongly underestimated in all the simulations. Noticeably, the impact of the model selection when

representing aerosol optical properties is higher than the use of different emission inventories. On the other hand, the influence of ARI and ACI has a little visible impact compared to the impact of the model used.

## 1 Introduction

The Fifth Assessment Report (AR5) of the Intergovernmental Panel of Climate Change (IPCC) ascribes to aerosol and clouds the large uncertainty in the estimation of changes in the Earth's energy budget. Atmospheric aerosols produce these changes by two different ways: influencing the Earth's radiation, the aerosol-radiation interactions (ARI); and modifying clouds and precipitation, the aerosol-clouds interactions (ACI), which also increase uncertainty due to cloud processes (Boucher et al., 2013).

ARI and ACI strongly depend on the optical properties of atmospheric aerosols, along with their atmospheric distribution and hygroscopicity; and their ability to act as cloud condensation nuclei (CCN) and ice nuclei (IN). All these properties are highly variable in space and time scales due to aerosol particles' short-lived, non-uniform emissions, and the dependence of sinks on meteorology (Randall et al., 2007). Thus, the determination of atmospheric aerosol properties, by a complex interplay between their sources, atmospheric transformation processes and their removal from the atmosphere (Boucher et al., 2013) plays a part in the large uncertainty of aerosol effects on the Earth's climate.

It was in the 1980s when the atmospheric science community began to pay increasing attention to the atmospheric aerosol subject (Fuzzi et al., 2015). Since then, major efforts have been made to acquire better knowledge of atmospheric aerosol properties and their interactions with the Earth's climate to reduce the above-mentioned large uncertainty. Many regional field measurement campaigns have taken place; e.g., the Integrated Campaign for Aerosols, Gases and Radiation Budget (Moorthy et al., 2008, ICARB); the Megacity Impact on Regional and Global Environments field experiment (Paredes-Miranda et al., 2009, MILAGRO); the Integrated Project on Aerosol Cloud Climate and Air Quality interactions (Kulmala et al., 2011, EU-CAARI); Aerosol, Radiation, and Cloud Processes affecting Arctic Climate (Warneke et al., 2010, ARCPAC); among many others (Boucher et al., 2013). Moreover, global long-term aerosol measurements are taken by surface networks, such as Global Atmosphere Watch (Ogren, 2011, GAW), Aerosol Robotic Network (Holben et al., 1998, AERONET), the European Monitoring and Evaluation Programme (Tørseth et al., 2012, EMEP) or by satellite sensors, such as the Moderate Resolution Imaging Spectroradiometer (Remer et al., 2005, MODIS) or the Cloud-Aerosol Lidar and Infrared Pathfinder Satellite Observation (Winker et al., 2003, CALIPSO) among many other base measurements, base networks and instruments on board satellites (Boucher et al., 2013).

Measurements provide incomplete sampling but can be combined with information from global and regional aerosol models. There is a large number of international initiatives that study, among other climate issues, atmospheric aerosols and their climatic effects. Some examples are the Aerosol Comparisons between Observations and Models project, now in their Phase II (Schulz et al., 2009, AEROCOM-II), the Coupled Model Intercomparison Project, now in Phase 6 (Eyring et al., 2016, CMIP6), the Chemistry-Climate Model Initiative (Eyring et al., 2013, CCMI) or the Aerosol and Chemistry Model Intercomparison Project (Collins et al., 2016, AerChemMIP). Among these initiatives, the primary purpose of the Air Quality Model Evaluation

International Initiative (Rao et al., 2011, AQMEII) is to coordinate international efforts in scientific research on regional air quality model evaluations across the modelling communities of North America and Europe.

AQMEII Phase 1 (Galmarini et al., 2012) focused on developing general model-to-model and model-to-observation evaluation methodologies; while Phase 2 (Galmarini et al., 2015) focused on simulating aerosol/climate feedbacks with online coupled modelling systems. As part of this Phase 2, some studies evaluated aerosol properties and their effects on the climate system. Balzarini et al. (2015) analysed online model sensitivity to the chemical mechanisms of WRF-Chem chemistry-meteorology coupled model when reproducing aerosol properties; results found that although different chemical mechanisms gave different Aerosol Optical Depths (AOD), it was commonly underestimated. Forkel et al. (2015) found pronounced feedback effects, such as a reduction in seasonal mean solar radiation of 20 $Wm^{-3}$ and temperature of 0.25° in the summer of 2010 when ARI were considered. High aerosol concentrations resulted in a 10-30% decreased precipitation and low concentrations in very low cloud droplet numbers (5-100 droplets $cm^{-1}$) and a 50-70% lower cloud liquid water, which led to an increase in downward solar radiation of almost 50% when ACI were taken into account. Makar et al. (2015) evaluated the effect on chemistry due to the feedback between aerosols and meteorology. In this study, ACI were usually found to have a strong effect on ozone, particulate matter and other species, and also on the atmospheric transport and chemistry of large emitting sources such as plumes from forest fires and large cities. A similar work is that of Wang et al. (2015), in which a multi-model assessment of major column abundances of gases, radiation, aerosol and cloud variables was made using available satellite data. The evaluation results showed an excellent agreement between all the simulations and satellite-derived radiation variables, as well as precipitable water vapor. Other aerosol-/cloud-related variables, such as AOD, cloud optical thickness, cloud liquid water path, CCN and cloud droplet number concentration were moderately to largely underestimated by most simulations due to underestimations in aerosol loadings (Baró et al., 2018). These authors also highlighted the large uncertainties associated with current model treatments of ACI.

Moreover, and through the AQMEII Phase 2, the working group 2 of the COST Action ES1004 EuMetChem (European framework for online integrated air quality and meteorology modelling, http://www.eumetchem.info/) investigated the importance of different processes and feedbacks in online coupled chemistry-meteorology/climate models for air quality simulations and weather predictions. As part of this initiative, an important aerosol load episode, the Russian wildfires in 2010, was investigated. Results indicated that the inclusion of ARI led to a drop between 10 and 100 $Wm^{-2}$ in the average downward shortwave radiation on the ground and an almost 1° in the mean temperature (Forkel et al., 2016; Toll et al., 2015a). During the same episode, Baró et al. (2017) found a reduction in the 10-meter wind speed of 0.2 $ms^{-1}$ (10%) because the presence of biomass burning aerosols implied a reduction in shortwave downwelling radiation on the surface which, in turn, led to a reduction in the 2-meter temperature. Thus, it led to a reduction in the turbulence flux, and developed a stabler planetary boundary layer. Kong et al. (2015) and Palacios-Peña et al. (2017) evaluated the effects of the inclusion of ARI and ACI for this 2010 wildfires episode and a desert dust outbreak. These results showed that a minor, but significant, improvement was observed when ARI and ACI were taken into account.

AQMEII Phase 3, to which this work contributes, focused on evaluating and intercomparing regional and coupled global/regional modelling systems by collaborating with the Task Force on Hemispheric Transport of Air Pollution, Phase 2 (Dentener et al.,

2015, HTAP2). The simulation strategy followed the proccedure adopted in the first two AQMEII Phases, as described in Galmarini et al. (2012, 2015, 2017).

On the other hand, several previous studies evaluated modelled aerosol optical properties against satellite data from a global point of view. In Ghan et al. (2001), simulated AOD were within a factor of 2 with respect to AVHRR (Advanced Very High Resolution Radiometer) products and the behaviour of the Ångström Exponent (AE), estimated from POLDER (POLarization and Directionality of the Earth's Reflectances) and SeaWiFS (Sea-Viewing Wide Field-of-View Sensor), was similar to that simulated. Otherwise, both the simulated AOD in Chin et al. (2002) and Reddy et al. (2005) were reproduced with most of the notable features in TOMS (Total Ozone Mapping Spectrometer), AVHRR and MODIS. Moreover, Ginoux et al. (2006) revealed sensitivity to humidity when evaluating modelling results against satellite data. Kinne et al. (2003) compared aerosol modules from seven models with MODIS and TOMS, and found large discrepancies over tropical and Southern Hemisphere oceans due to the sea salt treatment. Kinne et al. (2006) also discovered a lower simulated AOD among 20 different modules from the AEROCOM Project (0.11 to 0.14) when comparing simulations with the satellite AOD composite of MODIS, MISR (Operational Microwave Integrated Retrieval System), AVHRR, TOMS, and POLDER retrievals (0.15).

More recent studies are Colarco et al. (2010), who assessed simulated AOD *versus* MODIS and MIRS, and found similar seasonal and regional variability and magnitude over downwinds of the Saharan dust plume, a high bias in sulphate-dominated regions of North America and Europe, and a better agreement over ocean when the sea salt burden was reduced by a factor of 2. Furthermore, Zhang et al. (2012) reported a relative difference in AE of 13.8% with a negative(positive) bias over high latitude regions(oceans), but a good correlation for AOD in comparison with MODIS. Finally, Liu et al. (2012) evaluated long-term simulations compared with the satellite composite derived by Kinne et al. (2006) and identified a low bias for AOD, but a good representation of the observed geographical and temporal variations of aerosol optical properties.

Similar studies to the one presented in this contribution are those of Jeuken et al. (2001), who made a seasonal comparison (over Europe) of AOD calculations with ATSR-2 (Along Track Scanning Radiometer 2) on board the European ERS-2 satellite. The results showed an average difference of 0.17-0.19, but a good representation of the observed patterns. Simulated AOD in Solmon et al. (2006) presented a general underestimation (more pronounced over the Mediterranean Basin), but within the range of AERONET and MIRS over northern Europe, and common spatial patterns to those of MODIS and TOMS over both Europe and Africa.

Recently, Curci et al. (2017) used AQMEII Phase 3 simulations to evaluate black carbon absorption against AERONET; but no works have evaluated the modelled seasonal representation of optical properties against satellite observations over Europe with the variety of regional models involved in AQMEII Phase 3. This represents an added value of the current contribution because: 1) all the regional models evaluated here were run using the same boundary and initial conditions, which permits us to investigate the importance of different processes and feedbacks in each model; 2) the use of two different emissions datasets allows the evaluation of the influence of these emissions in the representation of aerosol optical properties; and 3) the use of online coupled chemistry-meteorology/climate models (as some of the models used here) permits the investigation of the influence of ARI and ACI. As above-mentioned, aerosol optical properties influence ARI and ACI, and hence a good representation of them is, thus, a key issue to reduce the uncertainty of aerosol effects on the Earth's climate system. For this

reason, our main aim was to evaluate the representation of two fundamental aerosol optical properties, AOD and AE, using the models included in the AQMEII Phase 3 initiative over Europe. The evaluation was made by using remote-sensing observations from the MODIS sensor and from AERONET and MAN (Maritime Aerosol Network). Section 2 provides a brief description of the observations and models, and the evaluation methodology. Section 3 presents and discusses the evaluation results. Finally,

Section 4 summarises the main conclusions reached.

## 2   Methodology

In this work, we focused on evaluating the representation of aerosol optical properties (AOD and AE) over Europe throughout the year 2010. The evaluation was conducted using remote sensing data from the MODIS sensors onboard the Terra and Aqua satellites; and AERONET and the Maritime Aerosol Network (MAN) ground-based networks.

### 2.1   Model simulations

The evaluated simulation data were taken from the regional chemical-meteorology simulations made over Europe within the framework of the AQMEII Phase 3 initiative.

Two different anthropogenic emissions data were used. On the one hand, the HTAP_v2.2 (referred to from this point onwards as HTAP emissions). These data were harmonised by the Joint Research Centre's (JRC) Emission Data Base for Global Re-

search (EDGAR) team in collaboration with regional emission experts from different agencies from the United States, Europe and Asia. HTAP emissions covered the years 2008 and 2010, with yearly and monthly time resolutions, and a global geo-coverage with a spatial resolution of $0.1°$. The chemical species were $SO_2$, $NO_x$, NMVOC, $CH_4$, CO, $NH_3$, $PM_{10}$, $PM_{2.5}$, BC and OC at the sector-specific level. There were seven emission sectors included (air, ships, energy, industry, transport, residential and agriculture) (Janssens-Maenhout et al., 2015; Galmarini et al., 2017).

On the other hand, the MACC emissions (Pouliot et al., 2015) was used. MACC was previously used for AQMEII Phase 2 (Galmarini et al., 2015). The dataset is a follow-on to the widely used TNO-MACC database (Pouliot et al., 2012), with a base resolution of $\sim 7km$. The provided species were: $CH_4$, CO, $NO_x$, $SO_x$, NMVOC, $NH_3$, $PM_{coarse}$, $PM_{2.5}$. A separate PM bulk composition profile file was composed, based on information per source sector and per country. The different represented chemical components were EC, OC, $SO_4^{-2}$, sodium and other mineral components. For all the AQMEII Phase 3 participants,

wild fire emissions were included as in Pouliot et al. (2015) and Soares et al. (2015) but volcanic and dimethyl sulphide emission (DMS) were not considered (Galmarini et al., 2017).

The study period was 2010 and the target domain was Europe. A detailed description of the simulations can be found in Solazzo et al. (2017). However, a brief summary focused on aerosol treatment is provided below, and summarised in Table 1.

The FI1 simulations were run at the Finnish Meteorological Institute (FMI), and the only difference between both FI1 simu-

lations was the type of emissions used (HTAP or MACC). The System for Integrated modeLling of Atmospheric coMposition (SILAM), version 5.4. (Sofiev et al., 2015), was run with the meteorological input extracted from the European Centre for Medium-Range Weather Forecasts (ECMWF). Sea salt emissions were included as in Sofiev et al. (2011) (but not for bound-

aries), and biogenic volatile organic compounds (VOC) emissions were taken from Poupkou et al. (2010). Wind-blown dust was included only from lateral boundary conditions. Gas phase chemistry was simulated with Carbon-Bond Mechanism-IV (CBM-IV), and with updated reaction rates according to IUPAC (http://iupac.pole-ether.fr) and JPL (http://jpldataeval.jpl.nasa.gov) recommendations. Secondary inorganic aerosol (SIA) formation was computed with the updated DMAT scheme (Sofiev, 2000)

and secondary organic aerosol (SOA) formation with the Volatility Basis Set (Donahue et al., 2006, VBS). AOD in SILAM was calculated assuming external mixture of spherical particles, taking into account their hygroscopic growth. The optical properties used in the Mie computations come from the OPAC dataset (Hess et al., 1998).

The ES1 simulation was run by the Regional Atmospheric Modelling Group at the University of Murcia (UMU, Spain). They used the Weather Research Forecasting model online coupled with Chemistry (Grell et al., 2005, WRF-Chem), version

3.6.1. Meteorological inputs were driven by ECMWF analysis fields. The aerosol module was the Modal Aerosol Dynamics Model for Europe (Ackermann et al., 1998, MADE), in which secondary organic aerosols (SOA) were incorporated by using the Secondary Organic Aerosol Model (Schell et al., 2001, SORGAM). The gas phase chemistry mechanism was the Regional Acid Deposition Model, version 2 (Stockwell et al., 1990, RADM2), with 57 chemical species and 158 gas phase reactions, among which 21 are photolytic. Anthropogenic emissions were MACC emissions. Biogenic VOC emissions were computed

by applying the Model of Emissions of Gases and Aerosols from the Nature (MEGAN) emissions model (Guenther, 2006), version 2.04. The MADE/SORGAM sea salt (Gong, 2003) and dust (Shaw et al., 2008) emissions were used.

The IT1 simulation was conducted at Ricerca sul Sistema Energetico (RSE, Italy) using the Weather Research Forecasting (WRF) model coupled with the Comprehensive Air Quality Model with Extensions (CAMx), version 6.10. Meteorological inputs were generated using WRF version 3.4.1. Anthropogenic emissions were MACC and biogenic emissions were esti-

mated by MEGAN. WRF-Chem was adopted to predict GOCART (Goddard Chemistry Aerosol Radiation and Transport) dust emissions (Ginoux et al., 2001) along with meteorology. Sea salt emissions were computed using de Leeuw et al. (2000) and Gong (2003) methodologies. The WRF-CAMx pre-processor (ENVIRON, 2014, version 4.2) was used to create the CAMx ready input files by collapsing the 33 vertical layers used by WRF to 14 layers in CAMx, but maintaining the layers up to 230 m above ground level identical. Aerosol optical properties were estimated by means of the Aerosol Optical DEpth Module

(Landi, 2013, AODEM) post-processing tool that was coupled to CAMx regional model. AODEM calculated the optical properties (e.g. AOD, extinction and scattering coefficients, and particle number concentrations) at different wavelengths and size bins starting from the aerosol mass concentration predicted by CAMx. In this work, the Mie theory was applied by dividing the size range (40 nm to 10 $\mu$m) into 10 bins and calculating the hygroscopic growth of each aerosol species in each bin with the Hanel formula. Moreover, particles were assumed to be internally mixed.

The IT2 simulations were run at the University of L'Aquila (Italy) using WRF-Chem (Grell et al., 2005), version 3.6. The modified MADE/VBS aerosol scheme (Tuccella et al., 2015) was included in this version. This scheme is based on MADE to treat inorganic aerosols along with the VBS approach (Ahmadov et al., 2012). MADE/VBS allows a better representation of the SOA mass. The Regional Atmospheric Chemistry Mechanism - Earth System Research Laboratory (RACM-ESRL) gas phase chemical mechanism (Kim et al., 2009) was used. Anthropogenic emissions were MACC emissions, adapted to the chemical

mechanism used following the method of Tuccella et al. (2012). As for the IT1 and ES1 simulations, biogenic emissions were

calculated online by the MEGAN model (Guenther, 2006). Finally, the meteorological analyses used to initialise WRF were provided by the ECMWF with a horizontal resolution of 0.5° every 6 h. IT2_M-ARI was run with ARI, while large-scale clouds were solved by a simple module. IT2_M-ARI+ACI took into account ARI and ACI, while aqueous chemistry was solved in convective clouds. As for ES1, IT2 simulations used the MADE/SORGAM sea salt and dust emissions.

5    WRF-Chem simulations (ES1 and IT2) calculated aerosol optical properties according to wavelength following Fast et al. (2006), Chapman et al. (2009) and Barnard et al. (2010). The composite aerosol optical properties were determined by the Mie theory, adding over all size bins and wet particles diameters. An overall refractive index for a given size bin, as determined by a volume averaging, assuming an internal mixing, of complex indexes of refraction associated with each chemical constituent of the aerosol, was used. The inclusion of ACI and ARI in WRF-Chem is described in Chapman et al. (2009).

**Table 1.** Model simulations

| Model Code | Insti-tution | Meteorolo-gical model | Dispersion model | Emi-ssions | Aerosol mech. (dust sources) | AOD/AE estimation | Gas Phase mech. | Resolution (XY,Z)* |
|---|---|---|---|---|---|---|---|---|
| FI1_HTAP | FMI | ECMWF | SILAM v.5.4. | HTAP | DMAT-VBS (boundaries) | prognostic /diagnostic | CBM-IV | 0.25°, 12 uneven levs. below $13km$ ($1^{st}$ to $\sim 30m$) |
| FI1_MACC | | | | MACC | | | | |
| ES1_MACC | UMU | WRF | WRF-Chem v3.6.1 | MACC | MADE-Sorgam (online + boundaries) | prognostic /diagnostic | RADM2 | $23km$, 33 levs. up to 50hPa ($1^{st}$ to $\sim 21m$) |
| IT1_MACC | RSE | WRF v.3.4 | CAMx v6.10 | MACC | Coarse-Fine (online + boundaries) | diagnostic | CB05 | $23km$, 14 levs. up to $8km$ ($1^{st}$ to $\sim 25m$) |
| IT2_M-ARI | UAq | WRF | WRF-Chem v3.6 | MACC | (ARI) MADE/VBS (ARI+ACI) (online + boundaries) | prognostic /diagnostic | RACM-ESRL (Aq. conv. clouds) | $23km$, 33 levs. up to 50hPa 12 below $1km$ ($1^{st}$ to $\sim 12m$) |
| IT2_M-ARI+ACI | | | | | | | | |

FMI (Finnish Meteorological Institute, Finland), UMU (University of Murcia, Spain), RSE (Ricerca sul Sistema Energetico, Italy), UAq (University of LÁquila, Italy)

*XY: Horizontal resolution; Z: Vertical resolution.

10    A multimodel ensemble (henceforth referred to as ENSEMBLE) of the available simulations was also evaluated. The results presented herein did not intend to represent an ensemble of opportunity, but were merely calculated as the mean of all the

participating simulations. As part of the AQMEII Phase 3 initiative, the available variables of aerosol optical properties were AOD at 470, 550 and 675 nm.

## 2.2 Observational Data

The observational data used was obtained from the twin MODIS (Moderate Resolution Imaging Spectroradiometer) sensors. These instruments, aboard the Terra (MOD04_L2) and Aqua (MYD04_L2) satellites, provide information about aerosol optical properties around the world. Moreover, in order to conduct a reliable and complete analysis, we used ground-based observations from all the available stations in AERONET (Aerosol Robotic Network,https://aeronet.gsfc.nasa.gov/) and the available data from the MAN (Maritime Aerosol Network, https://aeronet.gsfc.nasa.gov/new_web/maritime_aerosol_network.html) which is a component of AERONET.

MODIS data came from Level 2 of the Atmospheric Aerosol Product (both MOD04_L2 and MYD04_L2) from the collection 6 (C6), with a resolution of 10 $km$. These data were estimated by two different algorithms, Dark Target (DT) and Deep Blue (DB). The used variables were: (1) a "combined" variable of the DT and DB algorithms which provide information about AOD at 550 nm for both ocean and land; and; (2) AE between 550 and 860 nm over the ocean estimated by the DT algorithm. There are several evaluations of this "combined" AOD products of MODIS C6 against AERONET sites around of the world (Sayer et al., 2014; Mhawish et al., 2017; Bilal et al., 2018). All of these established that a high percentage of retrievals are within the estimated error (EE) of the DT and DB algorithms, which is $(\pm 0.05 + 15\%)$ (Levy et al., 2013). Moreover, in Sayer et al. (2014) and Bilal et al. (2018) the performance of combined retrievals outperformed DT or DB retrievals in term of correlation (around 10%), meanwhile they showed relative mean bias values similar at a global scale. The preliminary estimated error (EE) for the used AE product was 0.45 in the pixels with an AOD > 0.2 (Levy et al., 2013). The selection of this observational data was based on results found by Palacios-Peña et al. (2018). These authors evaluated the uncertainty in the satellite estimates by comparing MODIS, OMI (Ozone Monitoring Instrument) and SeaWIFS (Sea-viewing Wide Field-of-view Sensor) AOD retrievals against AERONET observations. They found that MODIS presented the best agreement with the AERONET observations compared to other satellite AOD observations during two studies with high aerosol load during 2010 over Europe.

As Terra and Aqua are in Sun-synchronous orbits around the Earth, MODIS does not provide data over the entire studied domain for each time step. According to Levy et al. (2013), who have established that there is no significant difference between MODIS/AERONET comparability for Terra and Aqua data, we combined the hourly data from both satellites in order to obtain a whole year of data with a wider coverage for each time step than by using the Terra and Aqua data separately.

AOD at 675 $nm$ and AE between 440 and 870 $nm$ retrievals from AERONET Level 2.0 from the available European stations during the entire 2010 year were used. In the case of this network, the total uncertainty for the AOD data under cloud-free conditions is established as $< \pm 0.01$ for $\lambda > 440$ $nm$ and $< \pm 0.02$ for shorter wavelengths (Holben et al., 1998). The same variables were used from the Maritime Aerosol Network (MAN), which provided instantaneous ship-borne aerosol optical depth measurements. MAN estimated uncertainty of AOD in each channel is, as for AERONET, $< \pm 0.02$ because MAN is

affiliated with the AERONET calibration and data processing standards and procedures (Smirnov et al., 2009). Table 2 lists those time periods when the MAN data were available to our study.

**Table 2.** MAN period of measurements during the year 2010.

| Boat | JFM | AMJ | JAS | OND |
|---|---|---|---|---|
| Alliance | | | 20 Aug.-3 Sep. | |
| Ak Fedorov | | 06-10 May | | 23-24 Nov. |
| Ak Ioffe | | | 13-19 Sep. | |
| James Cook | | | | 17-18 Oct. |
| Oceania | | 8 Apr.-14 Jun. | 17-21 Aug. | |
| Polarstern | | 5-15 May | | 25 Oct.-8 Nov. |
| Zim Iberia | | 15-19 May | | |

## 2.3 Evaluation Method

Simulations (Table 1) and observed data had a different spatial resolution. Henceforth and beforehand, all the gridded data
(simulations and satellite) were preprocessed and bilinearly interpolated to a common working grid with a horizontal resolution of 0.25°.

As aforementioned, our objective was to evaluate the representation of the main aerosol optical properties: AOD and AE. Observed optical properties were not available in the same wavelengths as simulations. Thus, in order to evaluate AE from simulations, this variable had to be estimated through the Ångström empirical expression (Ångström, 1929, eq. 1), where $\lambda$ is
the wavelength and $\beta$ is Ångström's turbidity coefficient.

$$AOD = \beta\lambda^{-AE} \tag{1}$$

By partitioning equation 1 at two different wavelengths and taking algorithms, AE can be computed from the spectral AOD values (Eck et al., 1999, eq. 2). Hence it is possible to estimate AE between two known wavelengths, and to also use this AE to estimate AOD at other different wavelengths. However, as established in Ignatov et al. (1998), retrievals of AE under AOD
conditions lower than 0.1 are highly uncertain. For this reason, we chose the criteria to estimate AE over areas with AOD > 0.1. Moreover, and according to the EE for the AE products of MODIS, we set the AE values range between -0.5 and 4.0. It is widely known that AE values spread from 0 to 4 and even sometimes, when really coarse particles are presented, they can

reach negative values. Hence, we chose AE values between -0.5 as the lowest limit in order to cover possible negative values in a close smoothing value to the EE for the AE products of MODIS.

$$AE = -\frac{ln(\frac{AOD_{\lambda_2}}{AOD_{\lambda_1}})}{ln(\frac{\lambda_2}{\lambda_1})} \qquad (2)$$

All the observations used in this work are not provided temporally in a regular way. This means that the number of occur-
rences in each of the pixels for satellite data or in each AERONET stations were not the same. As the results in this work are shown as seasonal means and in order to show robust means estimated with a reasonable number of occurrences (as in Palacios-Peña et al. (2018)) a mask containing those pixels(stations) where the satellite(station) occurrences were higher than the 10% of the maximum possible occurrences, was implemented. The total coverage of accurate satellite products are limited by the application of different algorithms which apply physical theory and the mathematical procedures to convert the radiances
measured by the instruments to geophysical quantities (as the ones used in https://modis.gsfc.nasa.gov/data/). In this sense, the total number of accurate satellite products does not represent the total radiance measures, and therefore, the maximum of possible occurrences for satellite data was selected as the maximum of occurrences during each studied season (JFM, AMJ, JAS or OND) over the entire domain. On the other hand, AERONET provides long-term and continuous data, thus the maximum of possible occurrences was established as the maximum of solar-light hours, because of the use of sun photometers, in each
station during each season. Figure 1 shows the number of total observations and the number of observations used when the mask was implemented. This mask was not implemented in MAN data because this network portrays instantaneous data.

Once all the data had the same spatial and temporal resolution, and following Equation 2, the simulated AOD and AE were calculated at the observed wavelengths. Then the hourly data were evaluated using classical statistics such as: the mean of the individual model-prediction error or bias ($e_i$); the mean bias error (MBE); the mean of the absolute error (MAE); and the
coefficient of determination (r), according to Willmott et al. (1985), Weil et al. (1992) and Willmott and Matsuura (2005). It is widely known that AERONET and MAN provide punctual observations; thus, simulation values for the evaluation against these networks were extracted by using the nearest neighbour approach.

The MBE was estimated as in Equation 3, where $i$ represents each time step, $P$ is the simulation data and $O$ is the observational value. MBE provides an idea about the behaviour of the models, and indicates whether the model over- or underestimates
the variable measured by the satellite sensor.

$$MBE = n^{-1} \sum_{i=1}^{n} e_i = \overline{P_i - O_i} \qquad (3)$$

The MAE was calculated as in Equation 4 and provides an estimation of the magnitude of the error independently of over- or underestimation.

$$MAE = \langle n^{-1} \sum_{i=1}^{n} |e_i| \rangle = \overline{|P_i - O_i|} \qquad (4)$$

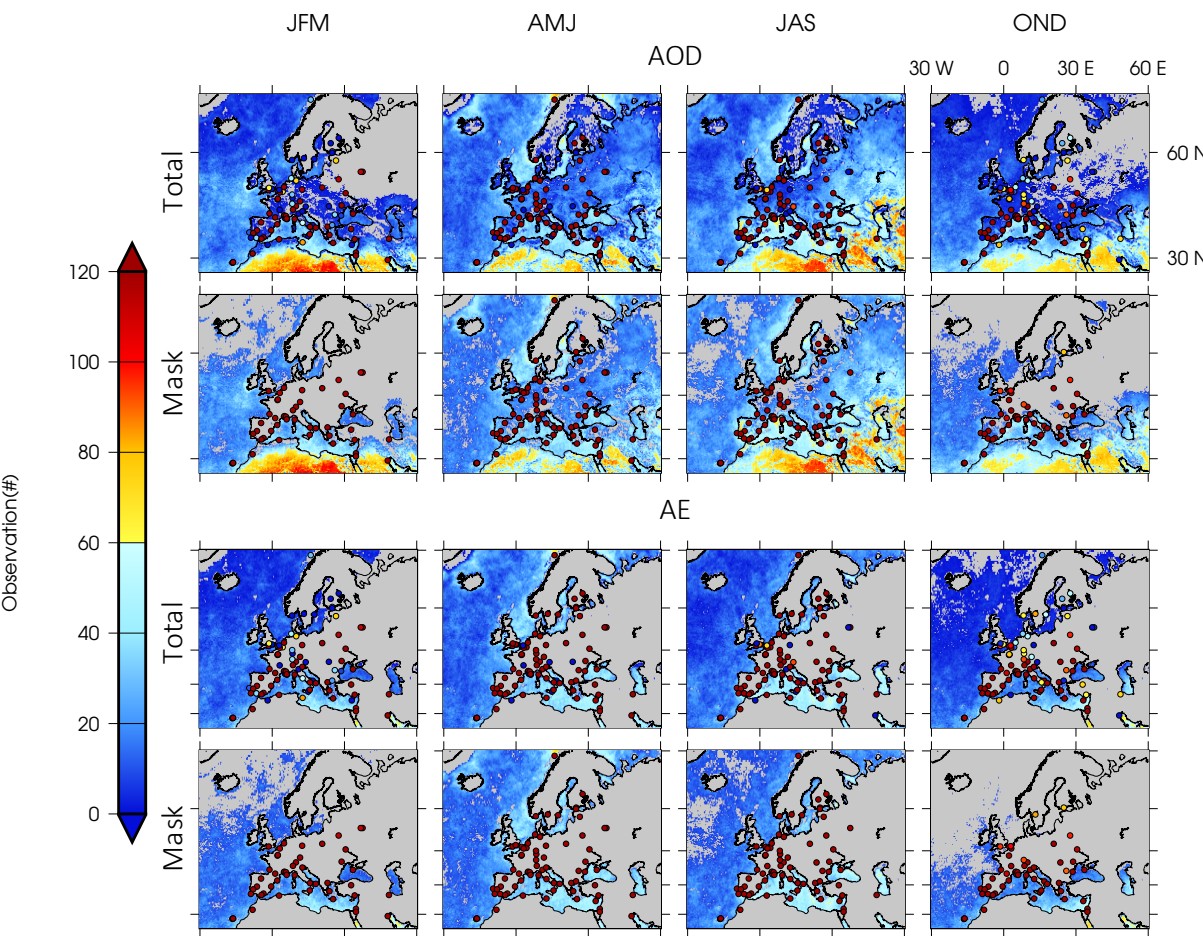

**Figure 1.** Total and under the mask number of observations used in the analysis. Maps show the number of MODIS observation and point the number of AERONET observations.

The temporal determination coefficient was estimated as in Equation 5 and was used as a measure of the strength of the linear relationship between two variables; in our case, the satellite and simulations values.

$$r^2 = \langle \frac{n^{-1} \sum_{i=1}^{n} (O_i - \overline{O})(P_i - \overline{P})}{\sigma_O \sigma_P} \rangle^2 \tag{5}$$

Finally, the Kernel probability density functions (PDF) with a broadband of 0.05 were used to evaluate the skills of the simulations to reproduce the spatio-temporal variability of the target variables (AOD and AE).

## 3 Results

This section evaluates the skills of the different AQMEII Phase 3 simulations regarding AOD and AE. The first section shows the model evaluation for AOD and the second for AE. The numerical result of each case for MODIS (M) and AERONET (A) are indicated by the numbers represented in each figure. Simultaneously, these results were listed in Tables S1, S2, S3 and S4 in the Supplementary Material. Finally, the skills of the simulations to reproduce the variability of AOD and AE are analysed using the PDF of each variable.

### 3.1 Model evaluation of the AOD representation

AOD is defined as the integrated extinction coefficient over a vertical atmospheric column and indicates to what degree aerosols avoid light transmission. AOD is not a direct function of the atmospheric load of particles, but can provide us an approximate idea of both atmospheric load of particles and the interaction of these particles with radiation.

Top row in Figures 2, 3 and 4 show the temporal means of AOD at 550 $nm$ values from a combination of the two MODIS satellites and of AOD at 675 $nm$ from AERONET stations. MAN data are displayed as diamonds linked by a colored line. Each color represents the track of each ship and accounts for instantaneous observations. The seasonal means and corresponding MAN data are presented in columns. JFM represents the temporal mean for January, February and March (from now on, winter); AMJ for April, May and June (spring); JAS for June, August and September (summer); and finally, OND for October, November and December (autumn).

When seasonal figures were analysed (Figure 2, 3 and 4), the highest values (around 1) were found over the southern part of the domain for all seasons due to frequent Saharan desert dust outbreaks, impacting the Mediterranean Region. Moreover, these desert dust outbreaks were more frequent and stronger for spring and summer over the southern part of the domain with mean AOD values above 0.4. In summer, the highest mean AOD values (above 1 for MODIS values) were found over a large area in Russia and its surrounding areas due to the heatwave and wildfires occurred over this area in 2010. However, in autumn and winter high mean AOD values were also found over the southern part of the domain, but were lower than 0.4. The lowest mean AOD value when considering space and time was found in autumn. It is noteworthy that AOD satellite values strongly agree with values by the available AERONET stations and MAN data. The gap in the satellite observations over the northern part of the land domain in winter and autumn is explained because ice, snow and clouds were avoided for the MODIS sensor, and aerosol properties were not retrieved over those areas (http://darktarget.gsfc.nasa.gov/). The gap in the rest of the seasons is explained by the limited number of observations (implemented mask explained in the Observational Data Section). Moreover, as the number of solar light hours is lower in the north during winter and autumn, this affected also the number of AERONET stations with available data. It explains the lack of AERONET data because the criterion applied (number of occurrences equal or higher 10% of the maximum of solar hours) was not met in a large number of AERONET stations. Throughout the seasons, high AOD values were obtained over the south-eastern part of the domain, outside Europe: Syria, Iraq, Kuwait and the Persian Gulf.

For the MBE, all the simulations presented (spatially) a similar behaviour in different seasons (Figure 2). The main common feature of the models was an overestimation of AOD over the southern part of the domain (the main area affected by desert dust outbreaks); and an underestimation -to a greater or lesser extent- over Russia, affected by the wildfire emissions in summer. It is remarkable that the results are similar when the evaluation is made against MODIS or AERONET. In certain occasions, mean spatio-temporal MBE was lower for the assessment versus AERONET data (may be because AERONET stations provided punctual data and could not be located near the areas with larger problems in the representation of AOD). MAN results could not be similar to MODIS and AERONET, since instantaneous data was used for the evaluation and not temporal means (as done for MODIS and AERONET).

As a general behaviour during all the seasons, FI1_HTAP and FI1_MACC, which use the ECMWF model for meteorology and SILAM for chemistry, showed a slight overestimation of AOD and higher AOD values than the rest of the models. This may be caused by the slower dry particle deposition in SILAM. This could explain that, although the size distribution is crudely represented, AOD is also very sensitive to this representation.

These values were overestimated over the southern part of the domain (northern part of the Saharan Desert), with values around 0.1. These values were spatially consistent with the higher MAE values (Figure 3). On the other hand, a slight under-estimation was found over the ocean areas probably caused by an underestimation of sea salt from boundary conditions. It should be noticed that no clear differences were found among simulations using HTAP and MACC emissions. All the other simulations (ES1_MACC, IT1_MACC, IT2_M-ARI and IT2_M-ARI+ACI) used the WRF meteorological model. When a different chemistry model was used, minor differences in the error were found between those simulations using the CAMx chemistry model (IT1_MACC) and WRF-Chem (IT2). These differences were of a similar order of magnitude to the differ-ences between the IT2 simulations including ARI and ACI. However, ES1_MACC (using WRF-Chem as the IT2 simulations) presented remarkable differences, by presenting a strong overestimation of AOD over the southern areas of the domain. This marked overestimation occurs because the dust scheme used in ES1_MACC lacked the gravitational settling. Although the IT2 simulations used the same dust scheme and model version, the dust flux was tuned in IT2 to estimate accurate dust concentra-tions. Hence, ES1_MACC showed the highest MBE and MAE values throughout the year when compared versus both MODIS and AERONET (see labels in Figures 2 and 3). IT1_MACC presented a general weak overestimation of AOD over the whole domain. IT2 simulations depicted a different behaviour. These simulations presented a general weak underestimation except over the southern part of the domain (areas affected by the Saharan dust outbreaks), where AOD was overestimated with low values. The IT1_MACC and the IT2 simulations presented the lowest absolute error values (see labels in Figures 2 and 3).

The ENSEMBLE notably overestimated the AOD values over the southern part of the domain, with very high AOD values over the northern part of the Saharan Desert. That is consistent with the high MAE values obtained. It should be highlighted that a noticeable underestimation was observed in the simulations over the south-eastern part of the domain (represented as a blue spot), centered over Iraq. The ES1_MACC simulation did not show this underestimation because of its high AOD values, but presented lower overestimation values (close to 0) over this area than over its surroundings. This small spot was also found for MAE (Figure 3). This can be explained by the fact that the emissions inventories used herein only covered Europe (see the Figure S5 in the Supplementary Material), thus the emissions over that area were not considered. Moreover, all the simulations

throughout the seasons overestimated the AOD over the southern part of the domain. This was related mainly to the high dust concentrations for to the boundary conditions; Solazzo et al. (2017) found that the error in primary species as dust was strongly affected by the emissions and boundary conditions in the AQMEII Phase 3 simulations.

In JFM (first column in Figures 2 and 3) all the simulations presented a weak underestimation over the Atlantic Ocean, except for IT1_MACC, which presented a weak overestimation in the northern part of the target domain (MBE MODIS mean of 0.01). The above-mentioned blue spot was clearly defined over a small south-easterly area and was stronger during this season, even for the ES1_MACC simulation with negative MBE values. For the IT1_MACC, IT2_M-ARI and IT2_M-ARI+ACI simulations, the highest MAE values were consistent over the latter area. FI1 simulations presented an overestimation of AOD over North Africa. This area was larger and with a stronger overestimation for ES1_MACC (MBE MODIS mean of 0.23 and AERONET mean of 0.07) for the same reason explained above. The ENSEMBLE presented an intermediate behaviour, with milder MBE and MAE values (0.02 and 0.12, respectively for MODIS; and 0 and 0.06 for AERONET): an overestimation of the AOD values over North Africa, a very weak underestimation over the Atlantic Ocean and the blue spot centered over Iraq and Syria.

In AMJ (second column in Figures 2 and 3), the underestimation of AOD was similar to that in winter, but with steeper values. All the simulations presented an overestimation (with different degrees) over the southern part of the domain, the Balkan Peninsula and southern Russia. This overestimation was larger and stronger for the ES1_MACC simulation (MBE MODIS mean of 0.21 and AERONET mean of 0.13) and once again presented higher MAE values (0.29, MODIS; 0.19, AERONET). All the simulations, except IT1_MACC, presented a weak underestimation over the Atlantic Ocean. The IT simulations gave lower errors than the rest. As in winter, a small south-easterly area (the blue spot) appeared, but was consistent with the maximum MAE values for the IT simulations.

The underestimation of AOD due to the wildfire emissions over Russia and the surrounding areas was one of the most important issues in JAS (third column in Figures 2 and 3). This underestimation was larger and stronger for the IT2 simulations, and was smaller and weaker for the FI1 simulations. Moreover, the aforementioned small area in the south-eastern part of the domain presented higher underestimation values over a larger area than during the other seasons, and reached as far as the Persian Gulf. Conversely, the FI1 simulations presented higher values and the IT2 simulations gave lower values. While the overestimation was stronger and affected a larger area than during any other season, this time the higher overestimation values were found over the northwestern areas of Africa and the Iberian Peninsula. As for the seasons, the ES1_MACC simulation showed the strongest and largest overestimation. During this season, with higher AOD values, all the simulations presented the highest error values.

The ENSEMBLE is conditioned by the most remarkable behaviour of MBE and MAE in the individual simulations. For example, if a simulation presents a strong underestimation (or overestimation) in a certain area, that is going to impact seriously the ENSEMBLE evaluation results. This is evident over those areas where other individual simulations presented a characteristic skill (mainly the south-western part of the domain, or Russia and the surrounding areas).

In this latter case (Russia area) two possible hypotheses, associated to an inaccurate representation of fires emissions, could explain this underestimation. As established in Palacios-Peña et al. (2018), this underestimation of AOD may be due to a

misinterpretation of aerosol vertical profile. In this sense, Soares et al. (2015) found an understated injection height of the total biomass burning emissions. A different hypothesis ascribes this underestimation purely to underestimated emissions. Toll et al. (2015b) found that while the daytime plumes from large fires were indeed lifted higher, the night time emissions and emissions from small fires were injected closer to the ground, making the average smoke transport distance even smaller than for the fixed emission height. Also, Soares et al. (2015) points out, after Wooster et al. (2005), that MODIS is not sensitive enough to register the fire radiative power of small or smoldering fires, and thus large fraction of those is missed in the emission data, including also strongly emitting peat fires. The 2010 Russian fires included some huge fires, but also numerous small ones over large areas, and a large fraction of those was probably missed by MODIS.

OND was the season with the lowest error values (close to zero in most of the domain). All the simulations showed overestimations close to the southern boundary and underestimations over Tunisia and Algeria. Both the overestimation and underestimation were lower for the IT simulations than for FI1. ES1_MACC was the only simulation with a different behaviour during this season with a high AOD overestimation over almost all the domain (0.25, MODIS; 0.10, AERONET).

The coefficient of determination ($r^2$) (Figure 4) was higher than 0.5 over most of the domain when comparing the simulation results against MODIS. In JFM, the highest $r^2$ values (around 1.0) were found over the north-eastern part of the African continent. In AMJ, these high values were found over central and eastern parts of Europe and North Africa. In JAS, the highest $r^2$ values correspond to Russia and its surrounding areas, and a part of the Atlantic Ocean in the south-western part of the domain. Finally, in OND, $r^2$ was lower than for the other seasons, especially over the Mediterranean sea and the Atlantic Ocean.

## 3.2  Model evaluation of the AE representation

AE indicates the relationship between the size of the particles suspended in the atmosphere and the wavelength of the incident light, and although there is not a direct correspondence between aerosol size and AE, this exponent provides an idea of the size of particles. Low AE values are related to coarse particles, such as desert dust or sea salt, and high values are associated with fine particles, such as anthropogenic source particles or biomass burning. The AE values are usually between 0 (or even slightly negative in coarse mode aerosol episodes) and 4 (Boucher, 2015). AE data from simulations are less available than for AOD because some models did not provide AOD at different wavelengths and therefore it was not possible to estimate AE following the methodology established above.

Seasonal means of AE between 550 and 860 $nm$ satellite values (only estimated over the sea) and between 440 and 860 $nm$ from AERONET stations and MAN data, are showed in the first row in the AE figures (Figures 5, 6 and 7). Generally, through the different seasons, low AE values were found offshore, where sea salt particles (coarse) predominated. Over the Mediterranean coast near the Saharan desert, low values were found due to the frequent desert dust outbreaks. High values were observed over coastal areas and inland in central Europe due to fine anthropogenic emissions (e.g. on-road traffic). Moreover, these values became lower from inland to offshore. In JFM, (first column in the first row in the AE figures) and OND (fourth column), the lowest values were found over the Atlantic Ocean and the Mediterranean Sea. Similarly, high AE values (around 1.5) were found over coastal areas and inland in central Europe. In OND, a small area over the north of the Caspian Sea with

values of 2.5 was found. In AMJ, as represented in the second column in the first row in the AE figures, the AE values showed a narrow range between 1.0 and 1.5 over most of the domain. Some exceptions were found for values close to 0.5 near the African continent, and values close to 2.0 in northern Europe. It is noteworthy that low AE values (close to 0.5) were uniformly distributed in AMJ over the southern part of the domain, while in JAS (third column in the first row in the AE figures) the

lowest AE values (lower than 0.5) were found mainly over the southern Atlantic Ocean. Values between 2.0 and 2.5 were estimated over north-eastern coasts and over central and northern Europe and the north of the Black and Caspian Sea. As for AOD, AERONET stations and MAN data showed very similar values to MODIS. As AERONET stations are located over the continent, the temporal and spatial mean of the results provides higher values due to a higher influence of anthropogenic emissions.

On a broad view through the different seasons, FI1_HTAP (driven by ECMWF meteorological model and the SILAM chemistry model) underestimated the AE over most of the domain. This underestimation was higher over areas near European coasts and inland, where the observations showed values around 1.5. The general underestimation was lower over the south-western part of the domain, where AE observations were close to 0.5. This simulation also presented the highest MAE values. This model estimated larger-sized particles than those retrieved by observations. As aforementioned, SILAM crudely represents

the particles size distribution, which impacted the AE representation because it may have be centered the size distribution on particles with a larger diameter. Despite the results obtained for AOD representation evaluation (due to the lack of dust gravitational settling), ES1_MACC presented low error values (MBE and MAE) through the different seasons for AE. This could be explained by the high dust concentration over southern areas, resulting in low AE values and thus compensating the tendency for producing high $PM_{2.5}/PM_{10}$ ratios (Solazzo et al., 2012; Balzarini, 2013; Solazzo et al., 2014). A very low overestimation

was found over areas close to Africa, and a more noticeable underestimation was found over areas near the European coast and inland. The IT1_MACC simulation generally overestimated the AE values over the Atlantic Ocean and the Mediterranean Sea (areas with AE close to 0.5). Over the areas near the coast of central and northern Europe, where the observations gave values around 1.5, this simulation presented a smaller underestimation than the other simulations. The IT2_M-ARI+ACI simulation showed an overestimation over the Atlantic and Mediterranean coast near North Africa, and a weak underestimation over the

coasts of the North and Baltic Seas and inland over the AERONET available stations. IT_MACC (WRF coupled to CAMx) and both WRF-Chem simulations (ES1_MACC and IT2_M-ARI+ACI) underestimated high AE values and overestimated low AE values and thus, they underpredicted the variability of this variable, consistently with Palacios-Peña et al. (2017, 2018). On the other hand, Solazzo et al. (2012); Balzarini (2013); Solazzo et al. (2014) found a severe underestimation for $PM_{10}$ concentrations over Europe for WRF-CAMx and WRF-Chem models, which could explain the overestimation of low AE values.

Moreover, they also found an underestimation of $PM_{2.5}$ concentrations which could also explain the underestimation of high AE values, since simulated particles underestimate the variability of the size distribution. Finally, ENSEMBLE presented a noticeable underestimation of the AE values over the European coast (including the Mediterranean Sea) and inland, probably due to the strong underestimation provided by the FI1_HTAP simulation strongly conditioning the results of the ENSEMBLE. Very low overestimation values were obtained over the Atlantic Ocean and near African coasts in the southern Mediterranean

Sea. Moreover, ENSEMBLE and the other simulations presented a strong underestimation over the two small areas with AE

values around 2.5. It is noteworthy that the evaluation results for the "Polarstern" ship of MAN during OND showed negligible bias values.

JFM results are represented in the first column in Figures 5 and 6. The FI1_HTAP simulation generally showed an underestimation of the AE values (-0.30 MODIS and -0.46 AERONET), which was stronger over areas near the European coast and the available AERONET stations. ES1_MACC presented the lowest error values (0.14 MODIS and -0.32 AERONET for MBE; and 0.33 MODIS and 0.40 AERONET for MAE). This simulation slightly underestimated the AE values over the Atlantic Ocean and the Mediterranean Sea, and presented an underestimation over small areas close to the European coast and over AERONET stations. Both the IT1_MACC and IT2_M-ARI+ACI simulations gave a general overestimation over most of the domain. IT1_MACC showed a very slight underestimation of the AE close to the European coast, but this simulation had the highest error values due to the strong overestimation. However, IT2_M-ARI+ACI displayed a really low bias (temporal and spatial AERONET MBE of 0) when compared with the available AERONET stations. ENSEMBLE depicted a high overestimation over the North Atlantic and the Mediterranean Sea, and a small underestimation close to the European coast and inland.

The second column of Figures 5 and 6 shows the results obtained in AMJ. For this season, FI1_HTAP underestimated the AE values over most of the domain and presented the highest error values (MBE of -0.62 and MAE of 0.64 against MODIS). This underestimation was larger when this simulation is compared with AERONET stations (MBE of -0.99 and MAE of 0.99). ES1_MACC displayed an intermediate behaviour when compared to the other simulations, with a weak overestimation over the North Africa coast and a more noticeable underestimation over the northern part of the domain. Notwithstanding, ES1_MACC presented the lowest absolute error when compared with MODIS. For AERONET, the lowest values were found for IT2_M-ARI+ACI, but this is probably because this simulation was not evaluated for northerly stations. IT1_MACC overestimated the AE values over the Atlantic Ocean and the southern part of the domain, and underestimated AE in areas over the European coast. IT2_M-ARI+ACI overestimated the AE values over the Moroccan Atlantic coast and the southern Mediterranean Sea, but small areas of underestimation were found over the Azores Islands and the northern coast of France. Moreover, values over the European AERONET stations were underestimated. Finally, ENSEMBLE produced a general underestimation over most of the domain, for both MODIS and AERONET. The overestimation was produced mainly over an area that lies north of the British Isles and the south-eastern part of the domain, where satellite values came close to 0.5.

All the simulations run for JAS (third column in Figures 5 and 6) displayed similar skills as in AMJ. Generally speaking, FI1_HTAP underestimated the AE values and presented the highest errors. During this season, ES1_MACC showed a larger area of underestimation and a smaller are of overestimation, but with similar error values as in spring. The overestimation of IT1_MACC was weaker, but the underestimation was stronger and over a larger area over the North and Baltic Seas. IT2_M-ARI+ACI also produced an overestimation over most of the domain, but it was weaker than for AMJ. Notwithstanding, this simulation presented a small area of underestimation over the Baltic sea. However, ENSEMBLE displayed a general underestimation that lowered from inland to offshore.

In OND (fourth column in Figures 5 and 6), the behaviour of simulations was similar to that shown in winter. FI1_HTAP produced a general, but weaker underestimation than in AMJ and JAS. During this season, ES1_MACC produced a general

overestimation over the Atlantic Ocean and the Mediterranean sea and an underestimation over the inland AERONET stations, but once again it gave the lower error values. IT1_MACC and IT2_M-ARI+ACI overestimated AE values over most of the domain with similar MBE and MAE values but underestimated AE values in eastern AERONET stations. Finally, ENSEMBLE depicted a weak overestimation over the Atlantic Ocean and the Mediterranean Sea, and a slight underestimation over the Black, Caspian and Red seas. The values of all AERONET stations were also underestimated.

Figure 7 shows the results of the determination coefficient ($r^2$). FI1_HTAP and IT1_MACC showed relatively high values (around 0.5) over the Mediterranean Sea, but over this area, all the other simulations presented values above 0.25. However, $r^2$ values were low when simulations were compared with AERONET stations and MAN data. It was very difficult to find a clear pattern for the coefficient of determination. During each season FI1_HTAP showed the highest determination values and ENSEMBLE the lowest $r^2$.

## 3.3 Variability

A good approach to evaluate the spatial and temporal variability of a variable is the Probability Density Function (PDF). This represents the density of counts for each value of the variable. In order to study how the AQMEII Phase 3 simulations represented the variability of AOD and AE, the PDF of both variables for each studied season are shown in Figure 8. In that Figure, first left column corresponds to the PDF of AOD at 550nm, second column to AOD at 675nm, third column to AE between 550 and 860 nm and fourth to AE between 440 and 870nm. First row corresponds to winter (JFM), second to spring (AMJ), third to summer (JAS), and bottom row to autumn (OND). Observed values (MODIS in first and third column and AERONET in second and fourth) was represented by a black line; the ENSEMBLE by a red line; FI1 simulations by green dashed lines; ES1 by a yellow dashed line; IT1 by a cyan dashed line; IT2_M-ARI by a blue dashes line and finally IT2_M-ARI+ACI by a blue dotted line. Due to the small number of MAN occurrences these data are not shown in this section. PDF for MODIS and AERONET data were evaluated separately because they were not represented by the same variable and over the same space and time. However, they represented a similar behaviour regarding the comparison of the variability of the simulations against observations.

The PDFs of AOD for the data corresponding to winter (JFM), spring (AMJ) and autumn (OND) presented a similar behaviour, both for MODIS and AERONET. The observed values showed a high probability for low values (between 0 and 0.5). The PDF of IT1_MACC for these seasons was the most similar one to both observations dataset. For these three seasons, this was the simulation with a lower absolute error when the temporal standard deviation from the simulations was evaluated against observations, as we can see in the Supplementary Material. During autumn, AERONET data and their respective PDF from simulations are narrower than those from MODIS, so the AOD values, both observed and modelled, were lower over AERONET stations. In JFM and OND, FI1 and IT2 displayed analogous PDFs with the highest probability for the lower AOD simulated values than those observed. However, during AMJ, these four simulations gave almost equal PDFs. ES1_MACC showed a remarkable skill for the representation of AOD in all seasons when compared with AERONET. The PDFs for this simulation estimated higher probabilities for high AOD values than the other simulations and the observed values. This fact was again due to the above-mentioned lack of gravitational settling in the dust scheme used. For this reason, the probability

of low AOD values was lower than for the rest. This behaviour was not observed when this simulation was compared against AERONET values, when the ES1_MACC PDF was similar to the rest. ENSEMBLE displayed in JFM, AMJ and OND a high probability for lower AOD simulated values than those observed; but with ENSEMBLE, the probability for the higher AOD values was higher than for observations.

The PDFs of the AOD representation were different in JAS. For this season, both IT2 simulations depicted the closest behaviour to the observed PDF values. As seen in the Supplementary Material, these simulations displayed the lowest MAE compared with the observed standard deviation. All the simulations and ENSEMBLE in this season presented a higher probability for the high AOD values than for observations.

The third and fourth column of Figure 8 represents the PDFs for the MODIS and AERONET AE values, respectively. As

for AOD, JFM and OND presented similar PDFs to MODIS observed values. The observed MODIS AE values showed a high probability for the low AE values, around 0.5, and a low probability for the high AE values. For AERONET, winter and autumn PDFs displayed high probability values for low AE values, but these showed higher probability for higher values than those observed in MODIS. For AMJ and JAS, the PDFs for the MODIS observed values were tray-shaped, with a high probability for AE values between 0.5 and 1.5. As well as in OND, these PDFs displayed their highest probabilities for AE values around

1.5. AERONET stations in AMJ and JAS showed a probability which increased from AE values of 0 to 2, where the probability decreased.

The FI1_HTAP simulation depicted a high probability for very low AE values, between 0 and 0.6 for MODIS and 0 and 0.4 for AERONET. IT1_MACC gave similar PDFs for all the seasons, and a high probability was found for AE between 1 and 1.5. FI1_HTAP and IT1_MACC were the simulations with the narrowest PDFs, pointing out to a strong underestimation

of the observed variability of the AE (as was also indicated in the evaluation of the temporal standard deviation shown in the Supplementary Material). The PDFs for ES1_MACC, IT2_M-ARI+ACI and ENSEMBLE were wider than for the other simulations. ES1_MACC and IT2_M-ACI+ACI showed a higher probability for AE values from 0 to 2 for MODIS and 0 to 0.7 for AERONET. However, IT2_M-ARI+ACI showed a high probability for slightly higher AE values than in the case of ES1_MACC. The behaviour of the ENSEMBLE was similar in all seasons and showed a medium behaviour when compared

to the rest of the simulations. ENSEMBLE showed a high probability that ranged from 0 to 1.5 for MODIS and 0 and around 0.7 for AERONET. Notwithstanding, all the simulated PDFs were narrower than the PDF for the observed values, thus all the simulations underestimated the representation of the AE. This is observed in the Supplementary Material, where the estimation of the MBE of the standard deviation gave negative results for all the seasons and simulations. The low variability of the simulations inland is noticeable (over AERONET stations). Meanwhile during all the seasons AERONET displayed a PDF

between 0 and 2 AE values, the evaluated simulations displayed PDFs between 0 and 0.6 for AE, indicating that all the simulations displayed really low AE values inland.

## 4 Summary and Conclusions

Although AQMEII Phase 3 focuses on evaluating and intercomparing regional and linked global/regional modelling systems, an evaluation of the simulations against observations was necessary in order to have an educated guess on the skill of the simulations when representing aerosol optical properties. Solazzo et al. (2017) analysed the performance of models for different
meteorological variables and chemical species. In order to perform a more detailed analysis of the performance of the models, this work focused on evaluating the representation of aerosol optical properties by using AQMEII Phase 3 simulations and remote-sensing observations. The evaluation of these variables is important because they strongly influence ARI and ACI and, thus, influence the atmospheric aerosol effects on the climate system.

As the Mediterranean Region is frequently affected by Saharan desert dust outbreaks, and an area over Russia and the
surrounding during summer 2010 was affected by wildfires, those areas presented the highest AOD for the year 2010. These AOD values were similar in both databases used, MODIS and AERONET (and, when applicable, MAN). As the AOD evaluated from each database was available at a different wavelength, some aspects in the evaluation differed. This was because all the simulations provided AOD at 550 nm and they could be evaluated against MODIS data, but not all them had AOD at 675 nm available (as for FI1_MACC and IT2_M-ARI, which only provided AOD at 550nm). Thus, there were no means of estimating
AOD at 675 nm for assessing versus AERONET data. This fact resulted in a lower number of simulations but with more trustable values. As aforementioned, here the ENSEMBLE was merely calculated as the mean of all the available simulations; thus, ENSEMBLE results show a mean behavior of the available simulations and in any case, this ENSEMBLE has been built with at least four simulations for the sake of representativeness.

All the simulations presented similar AOD spatial patterns and provided a good representation of the low and mean AOD
values. Slight AOD underestimations for all the simulations were found over the Atlantic Ocean and, this fact could be attributed to an underestimation of the sea salt concentrations. Albeit different models in this work used different schemes to estimate sea salt emissions, this underestimation could be attributed to an underestimation in the boundary conditions, which are the same for all the simulations. On the other hand, a major underestimation occurred during the wildfires episode over Russia in summer 2010. This underestimation is common for all the simulations regardless the emission inventory, the model or the
configuration used. Differences when different emission inventories were used could be expected but, as above-mentioned, all the AQMEII Phase 3 simulations used the same fire emissions nevertheless. Thus, this AOD underestimations should be bonded with an underestimation of the fire emissions. Toll et al. (2015b) found that while the daytime plumes from large fires were indeed lifted higher, the nighttime emissions and emissions from small fires were injected closer to the ground, making the average smoke transport distance even smaller than for the fixed emission height. Also, Wooster et al. (2005) pointed out
that MODIS is not sensitive enough to register the fire radiative power of small or smoldering fires, and thus a large fraction of those is missed in the emission data, including also strongly emitting peat fires. The 2010 Russian fires included some huge fires, but also numerous small ones over large areas, and a large fraction of those was probably missed by MODIS. A different hypothesis, established in Palacios-Peña et al. (2018) ascribes this underestimation to a misrepresentation of the aerosol vertical

profile modelled and may, therefore, be caused by the AOD representation given the understated injection height of the total biomass burning emissions found by Soares et al. (2015).

Moreover, a large underestimation was produced for all the simulations, irrespectively of the used meteorological and chemical model or emissions inventory, over a small area in the south-eastern part of the domain (the aforementioned "blue spot").

This can be explained by the fact that both emissions inventories used herein only covered European areas (see the Emission Map in the Supplementary Material), thus the emissions over that area were not included in the simulations. In addition, the AOD over the southern part of the domain was overestimated and this is related mainly to the high dust concentrations in the boundary conditions. In line with this, Solazzo et al. (2017) found that the error in primary species as dust was strongly affected by the emissions and boundary conditions in the AQMEII Phase 3 simulations.

Overall, FI1 simulations, which used the ECMWF files as SILAM model input, presented negative or negligible MBE values. It means that these simulations estimated higher AOD values than the rest of the simulations. This fact is explained because SILAM is known to have slower dry particle deposition than other models. This could explain that, although the band quite crudely represented size distribution, AOD is also very sensitive to the particle size representation. Moreover, no differences between FI1 simulations (different emissions inventories) were found, pointing out to a low impact of the use of

different emission inventories on the AOD representation. This fact can be observed when FI1 simulations were compared with the rest of the simulations: differences are higher due when using different meteorological or chemistry models than when implementing different emission inventories. IT1 (WRF meteorological conditions for CAMx) depicted slight positive MBE values. This could be due to the aerosol size range (40 nm to 10 $\mu$m) used by the AODEM tool. This tool calculated the optical properties from the aerosol mass concentration predicted by CAMx. As found by Permadi et al. (2018), when these authors

applied the AODEM tool to WRF-CHIMERE outputs, an AOD underestimation due to the lack of coarse particles (above 10 $\mu$m) was expected. However, the use of mass concentration of aerosols with size range up to 10 $mu$m could produce a high impact on aerosol optical properties, because absorbing particles as black carbon predominate within this size range. Hence, this could produce an overestimation in the AOD estimation, as for IT1. ES1 (WRF-Chem) presented a high AOD overestimation due to the dust outbreaks. This marked overestimation took place because of the dust scheme lacked the gravitational settling.

Although IT2 simulations used the same dust scheme and model version, the dust flux was modified for these simulations to estimate accurate dust concentrations, and hence these simulations presented lower AOD values.

The added value when the IT2 simulations were analyzed was the inclusion of ARI in one of them (IT2_M-ARI) and the inclusion of ACI and aqueous chemistry in convective clouds in the other one (IT2_M-ARI+ACI). However, no important differences were observed for the AOD representation among them. Thus, the differences in the use of a different meteorolog-

ical and chemistry model were stronger than the implementation of aerosol radiative feedbacks. However, IT2_M-ARI+ACI showed more negative MBE values than IT2_M-ARI, indicating that the simulations which include a complex treatment of clouds displayed lower AOD values. Romakkaniemi et al. (2012) found a relationship between a reduction in the AOD and the CCN. The inclusion of the ACI produces a reduction in the CCN by the condensation kinetics of water during cloud droplet formation. This results in a reduction of the cloud droplet number, the cloud liquid water and, finally, an increase in downward

solar radiation, as also found by Forkel et al. (2015) when ACI were taken into account in the AQMEII Phase 2 simulations.

It is important to highlight that for all the simulations and seasons, the highest determination coefficient ($r^2$) values were obtained over the areas with mean AOD values (observed values between 0.5 and 1.0), which were approximately the areas with the lowest error values. Thus, all simulations were skillful for representing the seasonal mean AOD values. The use of an ensemble (defined as the mean of all the participant simulations) improved this statistical figure.

High AE values (indicating a strong presence of fine particles) were found near central European coasts and inland, probably influenced by anthropogenic emissions. Low AE values (coarse particles) were observed over the southern part of the domain, close to the Saharan desert and over the Atlantic Ocean. It was also noteworthy that the AE values over the Atlantic Ocean were generally much higher in spring and summer than in autumn and winter. This means that the aerosol particles over ocean areas and near the coast in warm months were apparently finer than in colder months. This might be related to two different
hypotheses: a) weaker winds in warm months or b) hygroscopic growth, which could be greater in cold months generally because of higher relative humidity (RH).

AE modelling skills were lower than for AOD (larger errors). The simulation run with the SILAM model and using the ECMWF meteorological inputs (FI1_HTAP) largely underestimated AE over most of the domain (when compared to MODIS and AERONET). Hence, this model estimated larger-sized particles than observations. As aforementioned, SILAM roughly
represents size distribution, which impacted the AE representation because size distribution centres on particles with a larger diameter. A different hypothesis could be ascribed to the use of different anthropogenic emissions, since FI1_HTAP is the only case when the HTAP emissions were used. The simulations using WRF coupled to CAMx model (IT1_MACC) and both WRF-Chem simulations (ES1_MACC and IT2_M-ARI+ACI) underestimated high AE values and overestimated low AE values. Thus, they underpredicted the variability of this variable. These results are similar to those established in Palacios-Peña
et al. (2017, 2018). On the other hand, Solazzo et al. (2012); Balzarini (2013); Solazzo et al. (2014) and Im et al. (2015) found a severe underestimation of $PM_{10}$ concentrations over Europe for WRF-CAMx and WRF-Chem models, which could explain the overestimation of low AE values. These authors also found an underestimation of $PM_{2.5}$ concentrations which could also explain the underestimation of high AE values since simulated particles underestimate the variability of the size. An interesting fact is found for ES1_MACC: despite the lack of dust gravitational settling, it presented the lowest bias for AE. This could be
explained by the high dust concentration over southern areas, resulting in low AE values and thus compensating the tendency for producing high $PM_{2.5}/PM_{10}$ ratios.

A clear difference was found in the use of a different meteorological model when the AE were evaluated. FI1_HTAP (ECMWF meteorological inputs) highly underestimated AE values, while the simulations which used WRF as meteorological driver produces an underestimation of high AE values and an overestimation of low values. This could be related with a
misrepresentation of the relativity humidity (RH) by each meteorological model and the strong influence of RH in the aerosol optical properties due to hygroscopic growth (Yoon and Kim, 2006; Altaratz et al., 2013; Palacios-Peña et al., 2017).

As AOD at different wavelengths were not available for several runs, it was not possible to apply the AE estimation method explained in section 2.3 for all the simulations. Moreover, the spatio-temporal coverage of the entire domain for AE simulations could be affected by the restrictions established in the same section regarding data quality, reducing the number of available

data for AE evaluation. However, those restrictions made AE values more trustable for model evaluation. ENSEMBLE was used as the mean of the available simulations for the different AE at different wavelengths.

No clear spatial pattern was found for the coefficient of determination of AE. One striking fact in this case was that using the mean of all the simulations as an ENSEMBLE simulation did not improve $r^2$ with respect to individual simulations. In fact, the worse $r^2$ results were found for ENSEMBLE, while the highest $r^2$ was presented for FI1_HTAP (despite its high underestimation of the AE values, FI1_HTAP skill in the temporal AE representation was good).

A good approach to evaluate the spatial and temporal variability of a variable is PDF. A wide PDF indicates high variability for the studied variable, and a narrow PDF points out to a low variability. For AOD representation, all the simulations presented a similar PDF to the observed values. The behaviour of all the simulations was similar in winter, spring and autumn; FI1 and IT2 presented higher probabilities for lower AOD values than those observed; ES1 presented higher probabilities for high AOD values than those observed due to the above-explained misrepresentation of the dust gravitational settling. Finally, IT1 presented the most skillful PDF, except during summer (best skills for IT2_M-ARI+ACI given the higher probability of obtaining AOD values around 0.5). Regarding IT2 simulations, although IT2_M-ARI+ACI had higher MBE values when compared with IT2_M-ARI, the former presented better skills in the representation of the variability of AOD. One general conclusion can be extracted from the analysis of the PDF of AE: for this variable, all the simulations in all the studied seasons underestimated spatio-temporal variability.

Summarising, the errors in all the simulations for AOD were lower than for AE. For AOD, low and mean values were well-represented, but high values presented larger errors. High AOD values were overestimated because of an overestimation in the dust boundary conditions. The high AOD values due to biomass burning were underestimated, which should be ascribed to an understated injection height of the total biomass burning emissions or directly to underestimated fire emissions. Other high AOD values were underestimated because the emissions which produced these high values were not considered. The errors in the AOD evidenced the strong influence of emissions and boundary conditions in the estimation of aerosol optical properties. Generally speaking, the skills to represent the variability of AOD were acceptable. For AE, the SILAM simulation underestimated the observed values; and the WRF-CAMx and WRF-Chem simulations were those with the best skills in the representation of this variable. Overall, for all the simulations, the variability of this variable was pervasively underestimated.

There was a high impact in the use of different physical and chemical mechanisms used by each model. Differences were found when ECMWF meteorological inputs were used by the SILAM model (more pronounced for AE), in contrast with the chemistry models which used WRF as meteorological model. This supports the conclusion that the evaluation of air quality models needs to be supported by the analysis of meteorological fields found by Solazzo et al. (2017). Regarding the aerosol optical properties there is a really high influence of the RH due to the hygroscopic growth (Yoon and Kim, 2006; Altaratz et al., 2013; Palacios-Peña et al., 2017). Differences between the model system used (combination of meteorology and chemistry) were higher than the differences due to the use of a different anthropogenic emission inventory or the inclusion of aerosol radiative feedback (ARI and ACI).

Regarding emission datasets, there is not a high influence when a different anthropogenic emission inventory was used. However, there was a high influence of the emissions of primary particles as biomass burning emissions, whose misrepre-

sentation produces an impact in aerosol optical properties. When the aerosol radiative feedbacks were taken into account, the inclusion of the ACI and more complex cloud processes resulted in lower AOD values (higher errors) but a better skill in the representation of the variability of this variable.

Henceforth, further studies are needed to improve the representation of aerosol optical properties, along with other properties such as atmospheric distribution, hygroscopicity, or the ability to act as cloud condensation nuclei and ice nuclei. The results presented here for the representation of aerosol properties can help improving the process-understanding of ARI and ACI. Also, aerosol effects on meteorology and climate could reduce (or, at least, help characterising) the uncertainty in the estimations of changes in the Earth's radiation budget due to aerosols and clouds.

## 5  Data availability

The outputs from the simulations can be obtained by emailing to *rbianconi@enviroware.com*. MODIS data are publicly available on the MODIS Atmosphere website (https://modis-atmos.gsfc.nasa.gov/MOD04_L2/acquiring.html).

*Author contributions.* LP-P and PJ-G wrote the manuscript, with contributions from all co-authors; each co-author was responsible for conducting the numerical simulations of his/her group and putting the information available. LP-P compiled all the experiments and did the statistical analysis, with the support of PJ-G

*Competing interests.* The authors declare that they have no conflict of interest.

*Acknowledgements.* This work was conducted under the support of the AQMEII/HTAP Phase III initiative. The authors acknowledge Project REPAIR-CGL2014-59677-R and ACEX-CGL2017-87921-R of the Spanish Ministry of Economy and Competitiveness and the FEDER European programme for support to conduct this research. L. Palacios-Peña acknowledges the FPU scholarship (ref. FPU14/05505) of the Spanish Ministry of Education, Culture and Sport. G. Curci and P. Tuccella thank the EuroMediterranean Center for Climate reserach (CMCC) for providing the computational resources. We also thank the researchers and their staff who have been involved in the MODIS datasets (NASA).

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

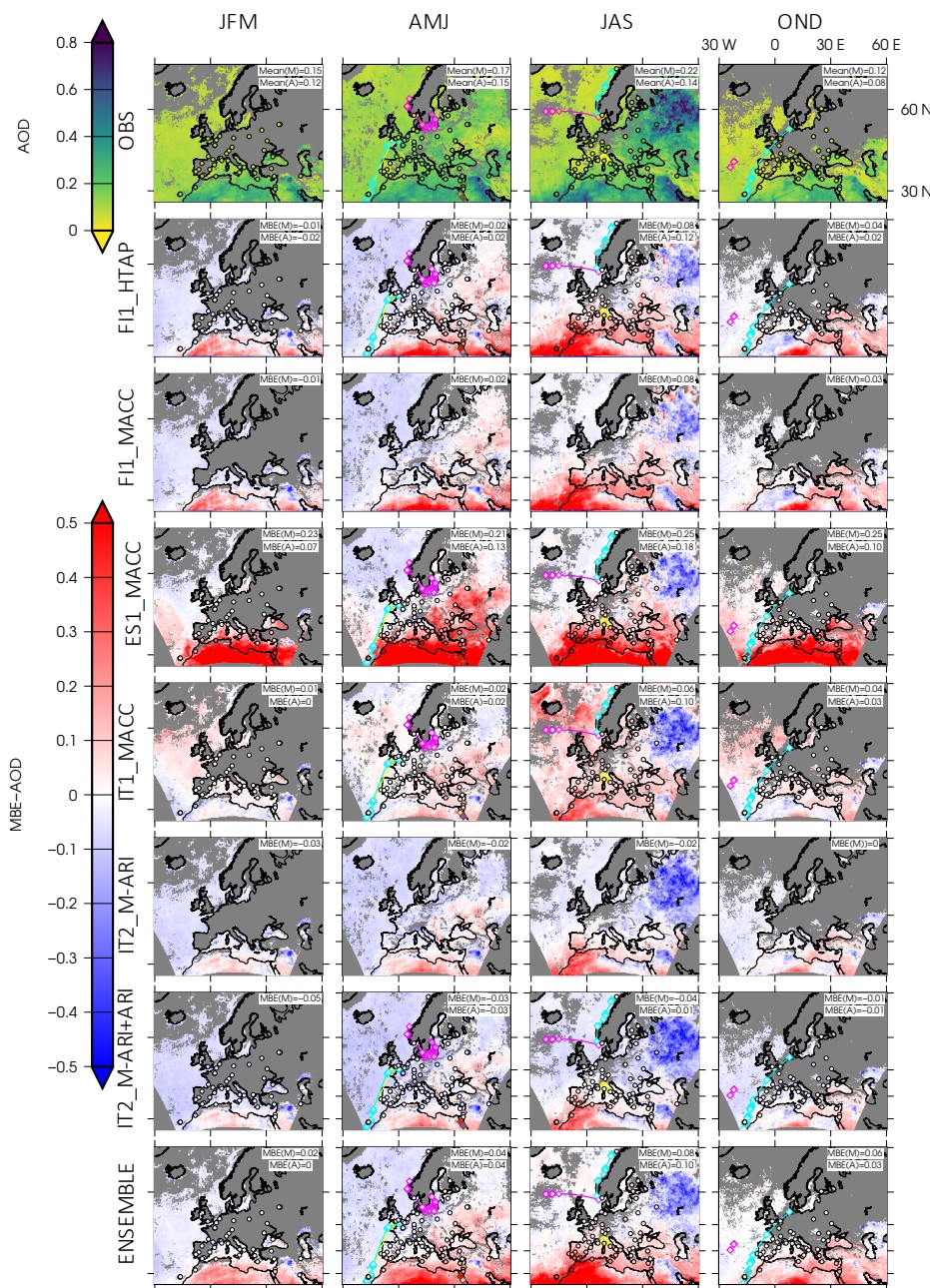

**Figure 2.** MBE results of AOD at 550nm from satellite (maps) and AOD at 675 nm from AERONET (points) and MAN (colored diamond) values *vs.* simulations at the same wavelengths. Columns from left to right, temporal mean of: winter (JFM), spring (AMJ), summer (JAS) and autumn (OND). First row: observations; and from second row to the bottom, MBE values of: FI1_HTAP, FI1_MACC, ES1_MACC, IT1_MACC, IT2_M-ARI, IT2_M-ARI+ACI and ENSEMBLE. Bonded lines between diamonds represent the ship track during AMJ: Ak Fedorov (yellow), Oceania (magenta), Polarstern (cyan) and Zim Iberia (chocolate); JAS: Alliance (yellow), Ak Ioffe (magenta) and Oceania (cyan); and OND: Ak Fedorov (yellow), James Cook (magenta) and Polartstern (cyan). Values in every plot indicate the spatial and temporal average of MBE for MODIS (MBE(M)) and AERONET (MBE(A)).[33]

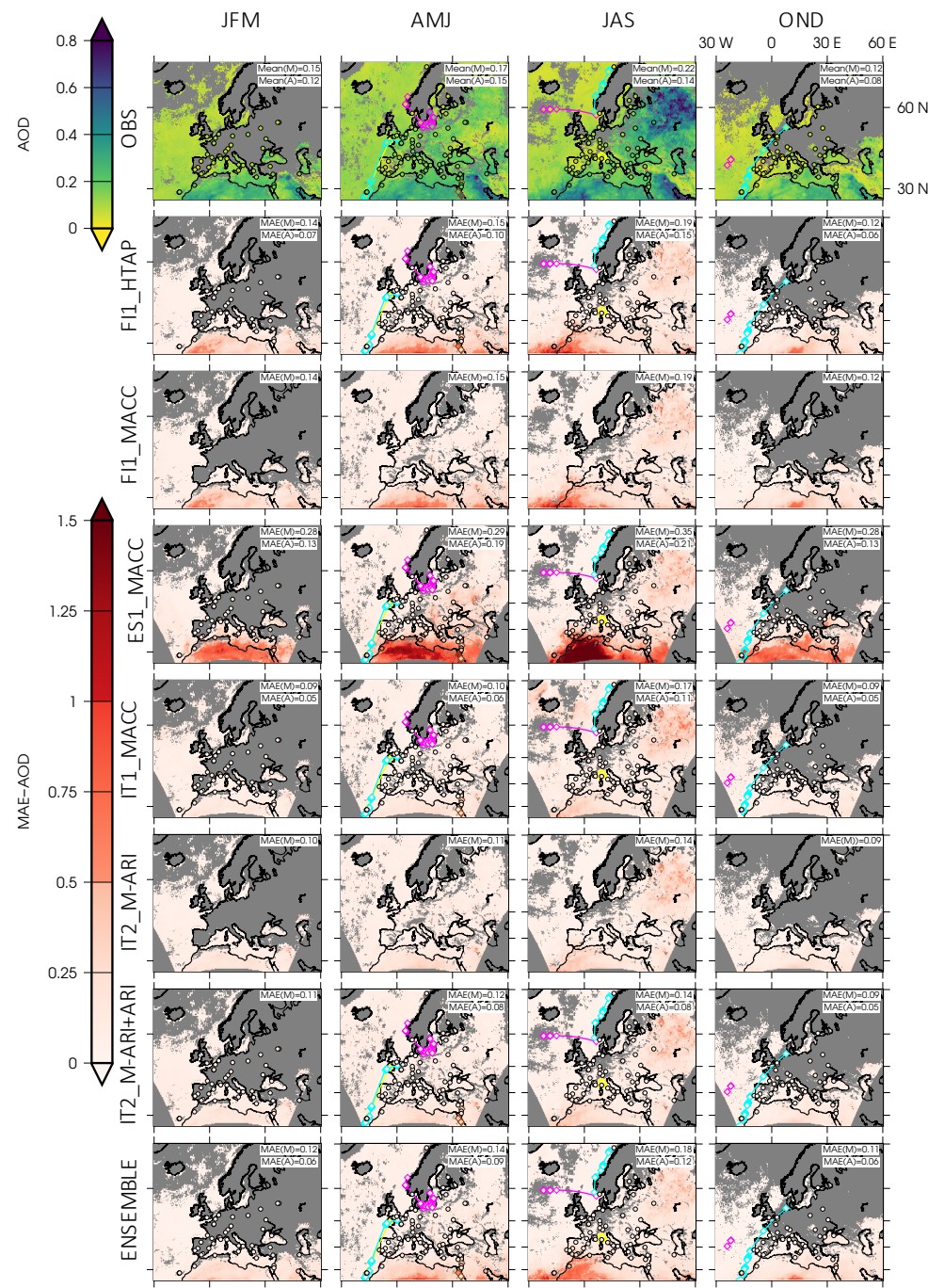

**Figure 3.** Id. Figure 2 for the MAE results.

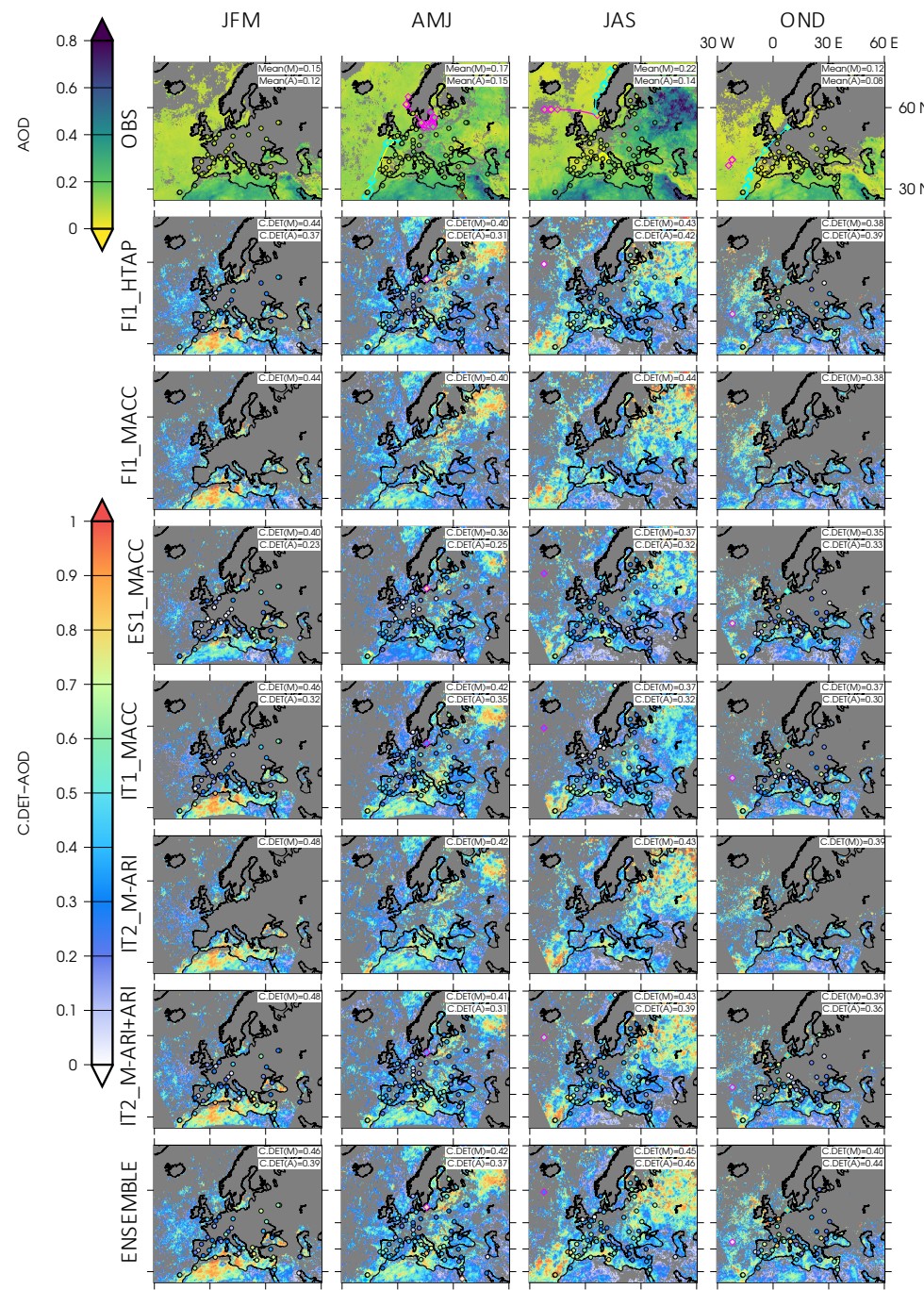

**Figure 4.** Id. Figure 2 for the determination coefficient.

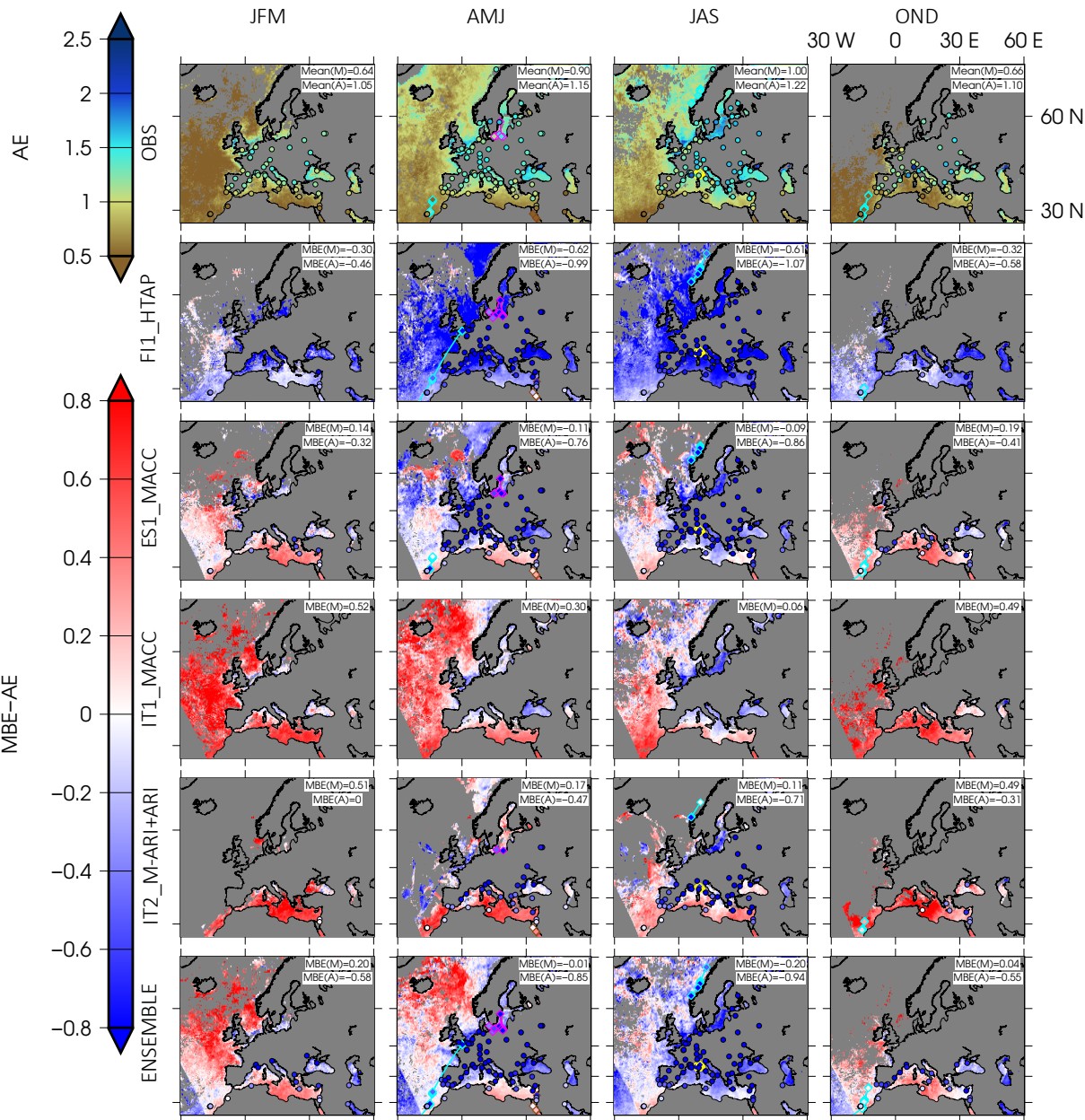

**Figure 5.** MBE results of AE between 550 and 860nm from satellite and AE between 440 and 870 nm from AERONET (points) and MAN (colored diamond) values *vs.* simulations at the same wavelengths. Columns from left to right, temporal mean of: winter (JFM), spring (AMJ), summer (JAS) and autumn (OND). First row: satellite values; and from the second row to the bottom, the MBE values of: FI1_HTAP, ES1_MACC, IT1_MACC, IT2_M-ARI+ACI and ENSEMBLE. Bonded lines between diamonds represent the boat track during AMJ: Oceania (magenta), Polarstern (cyan) and Zim Iberia(chocolate); JAS: Alliance (yellow) and Oceania (cyan); and OND: Polartstern (cyan). Values in every plot indicate the spatial and temporal average of MBE for MODIS (MBE(M)) and AERONET (MBE(A)).

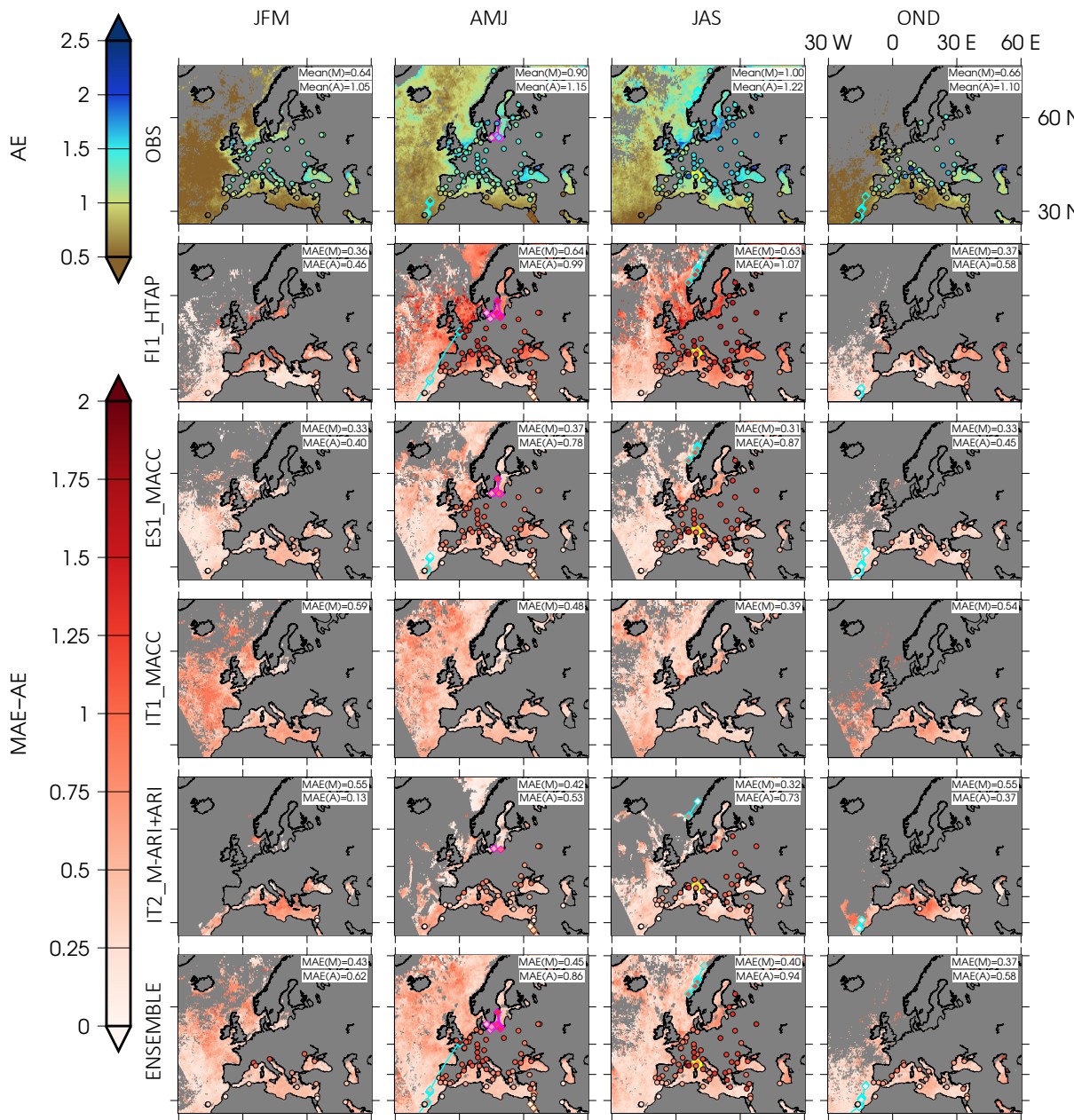

**Figure 6.** Idem Figure 5 for the MAE results.

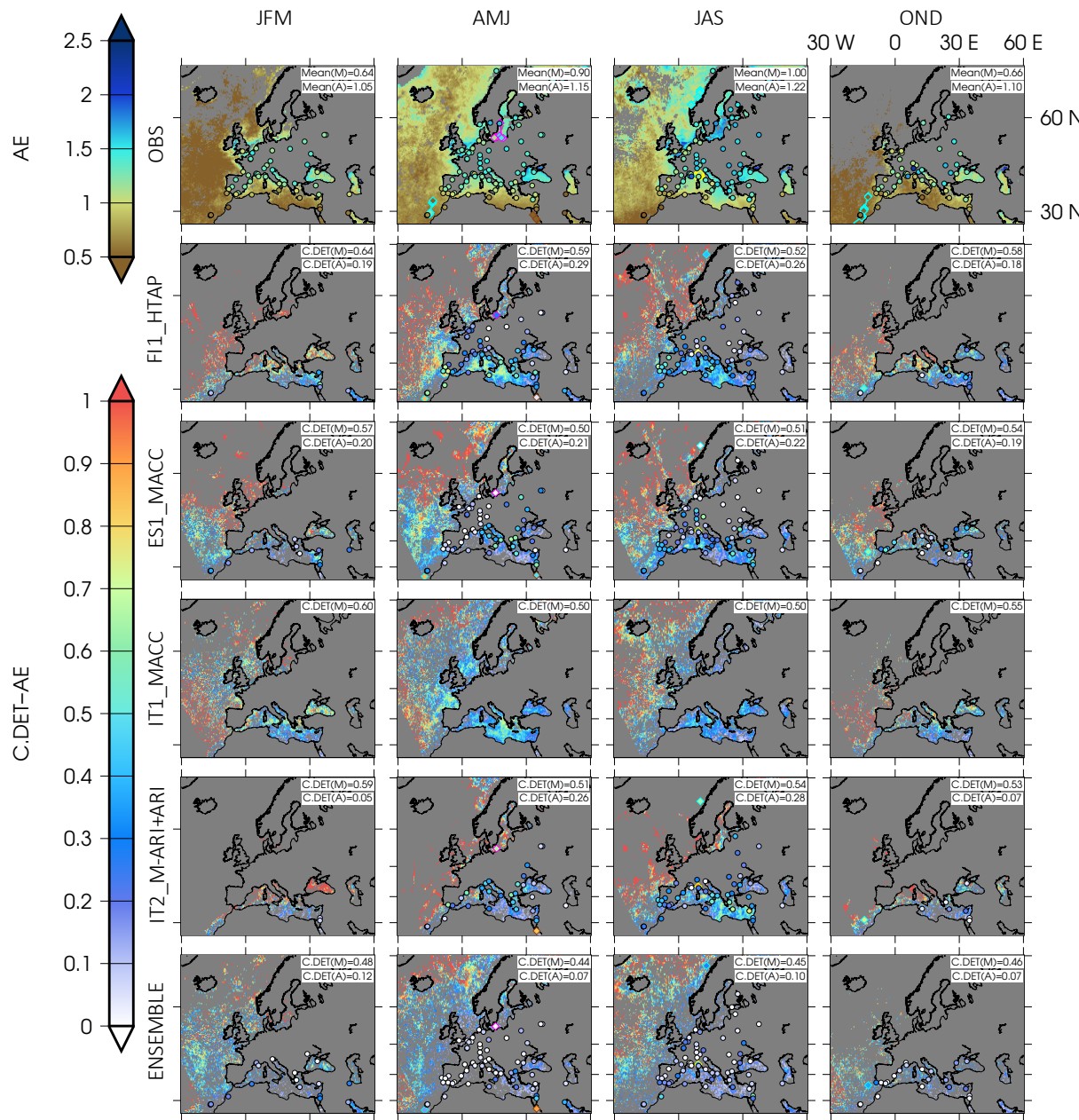

**Figure 7.** Id. Figure 5 for the determination coefficient.

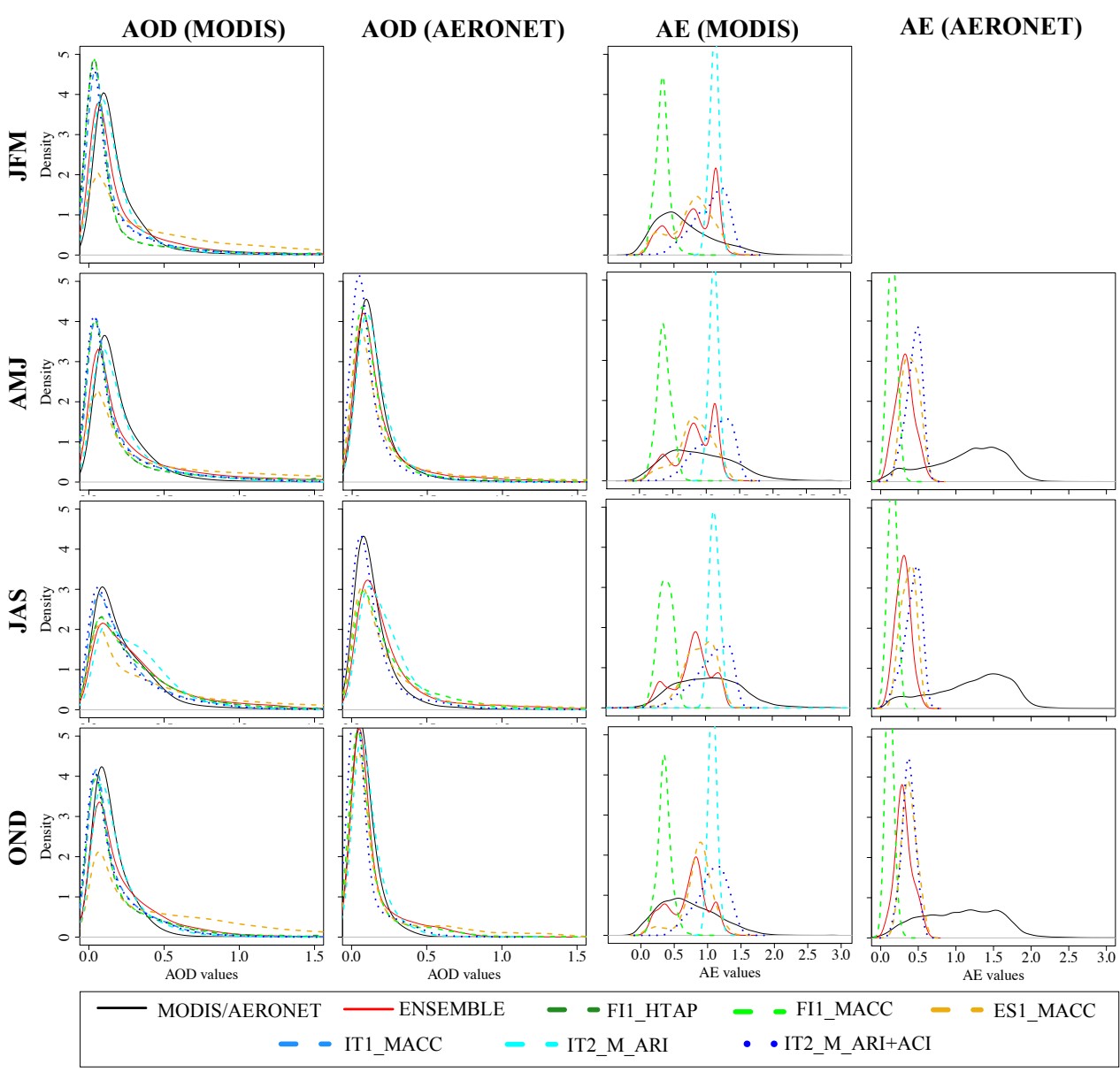

**Figure 8.** PDF for AOD (first, MODIS and second, AERONET columns) and AE (third, MODIS and fourth right, AERONET) values. From the top to the bottom: JFM, AMJ, JAS, OND.