# Peer review of "Aerosol optical properties over Europe: an evaluation of the AQMEII Phase 3 simulations against satellite observations"

_Atmospheric Chemistry and Physics, 2017_

## Referee Comment (RC1)

**Aerosol optical properties over Europe: an evaluation of the AQMEII Phase 3 simulations against satellite observations**
Laura Palacios-Peña, Pedro Jiménez-Guerrero, Rocío Baró, Alessandra Balzarini, Roberto Bianconi, Gabriele Curci, Tony Christian Landi, Guido Pirovano, Marje Prank, Angelo Riccio, Paolo Tuccella, and Stefano Galmarini

The authors present the evaluation of the aerosol optical properties simulated in the frame of the Air Quality Model Evaluation International Initiative. As a reference, MODIS AOD is used.
I have several concerns about the paper, which are specified below.
The manuscript needs a major revision before it can be considered for publication in ACP.

**General comments**

The language should be thoroughly checked.

The authors say that MODIS "combined" product is not validated (page 8, line 3). If this is true, I would be very careful to evaluate the model results with the product, which is not validated. However, this is not true. Several validation papers for MODIS C6 are published, among them:

http://www.mdpi.com/2072-4292/10/3/475

https://agupubs.onlinelibrary.wiley.com/doi/abs/10.1002/2014JD022453

https://www.atmos-meas-tech.net/6/2989/2013/amt-6-2989-2013.pdf

http://home.iitk.ac.in/~snt/pdf/Mhawish_RSE_2017.pdf

Please, include MODIS validation results and possible bias of the model results comparison, related to bias in MODIS AOD into the discussion.

MODIS provides very good but not perfect product, and validation results show weaknesses of the AOD product over certain areas. Thus, I strongly recommend to include AOD validation with AERONET (https://aeronet.gsfc.nasa.gov/) and MAN (https://aeronet.gsfc.nasa.gov/new_web/maritime_aerosol_network.html). Otherwise, the results of your analysis are biased by the MODIS quality.

MODIS coverage is surprisingly low in your analysis. How many (in %) pixels you discarded with your mask? Please, show it on the map. Please, repeat the analysis for all reported MODIS AOD pixels. As far as I know, MODIS deliver AOD data which passed the quality control.

**Specific comments**

Abstract.

- Please, add numbers (abs, or %) for under/overestimation results
- I disagree with the conclusion that "spatial and temporal variability of this variable is well-represented by all the models". Is that conclusion done from the comparison with MODIS? As I see from figure S3, almost all models have similar tendencies to underestimate, compared to MODIS, AOD over Siberia and underestimate AOD it the southern part of AOI

Page 4, Line 20. Please, correct to "ATSR"

Page 7, Line 14. Please, specify "x" in MxD

Page 8, Line 22. What is EE for AE MODIS product? Please, include the discussion to explain your choice for AE between -0.5 and 4.

Page 8, lines 4-5. Please, discuss briefly the results here.

Page 8, Line 27. Please, explain the mask in the other words. The current explanation is not clear. Was that mask applied for AOD? Or AE? Do you mean, that if for certain location max AOD was 1, you discarded all the cases when AOD<0.1 for that location? Why?
For AE, AOD limit of 0.1 is acceptable.

Page 8, Line 29. What is the measure of confidence here?

**Technical comments**

Please change the color scale for MODIS AOD to max. 1. With the current color scale, the AOD variation below 0.5 is hardly visible. I advice to include also the red color to the color panel.

---

## Referee Comment (RC2) · Anonymous Referee #3 · 16 Apr 2018

General comments

This is an interesting work that attempts to evaluate modeled AOD and AE over Europe using six model simulations performed under the AQMEII3 framework. The AQMEII initiative has provided a great opportunity for air quality model evaluation and model intercomparison across two continents that allows the community to assess the accuracy of the modeling systems, the drivers of their differences and make suggestions for future model improvements.

The main objection regarding the scientific methodology is the use of only one MODIS product to conduct model evaluation and intercomparison. The authors should take

advantage of all possible AOD/AE observations in the model domain over Europe, to enhance their understanding of model behavior. The paper needs grammatical editing to improve the language and some restructuring to improve the flow. There are a lot of clarifications needed for the methodology and discussion of the results. I am in favor of publishing this paper with Atmospheric Chemistry and Physics with Major Revisions. The specific comments that follow will help improve the discussion of the methodology and significance of the findings so that the overall quality of the manuscript is enhanced.

Specific comments

Abstract:

1. Line 13: "this variable" refers to AOD? Please be specific

2. General comment: It would be more beneficial to have a quantitative description of the conclusions. If AE is predicted more erroneously than AOD, add some quantitative measures to that statement in terms of errors/biases etc. Descriptive characterizations like "more serious errors" do not add any substantial information.

Introduction:

1. Line 1 and elsewhere: please refrain from using adjectives as "gravest". Keeping the discussion on scientific terms is more appropriate.

2. Lines 4-5 and elsewhere: please remove the phrase "so-called". The aerosol-cloud-radiation interactions are widely known; the definition of the acronym is enough.

3. General comment: The introduction includes a lot of information on past and recent work similar to this study. What is not clear is how this study is different from others. What is the new contribution made by this study to the scientific knowledge of modeled AOD and AE? A high-level brief statement would be appropriate in the introduction.

Methodology:

1. Page 6, IT1 simulation: "…Meteorological inputs were generated using WRF-Chem version 3.4.1. Anthropogenic emission were MACC and biogenic were computed by MEGAN.WRF-Chem was adopted to predict GOCART (Goddard Chemistry Aerosol Radiation and Transport) dust emissions (Ginoux et al., 2001) along with meteorology." What is the role of WRF-Chem in the WRF-CAMx simulations? It is not clear how those two modeling frameworks are combined. It doesn't make sense to use WRF-Chem for meteorology while WRF is already the modeling system used combined with CAMx.

2. Table 1 should include one more column that describes the origin of AOD/AE calculations: prognostic (i.e. during runtime) or diagnostic (i.e post-processing).

3. Description of dust sources for each simulation is very important since the domain covers North Africa and the signal from the satellite is much stronger there. Prescribed dust emissions (as total PM), online dust sources, etc. This information can be added to Table 1 under Aerosol Mechanism.

4. It's a surprise that the authors did not make use of AERONET data for such comprehensive AOD and AE model evaluation. This is strongly recommended for the revised version.

5. In addition, the merged AOD product has been used in a number of recent publications with one of the most important being Sayer et al. (2014) (https://agupubs.onlinelibrary.wiley.com/doi/pdf/10.1002/2014JD022453) which is a point of reference for the usage of these products.

6. Observational data: The authors state that "The selection of this observational data was due to results found by Palacios-Peña et al. (2017a)". More detailed explanation is needed for the selection of observational data than the reference to a previous publication. The readers should not have to read other papers to understand the basics of the approach.

7. Observational data (page 8, line 26): The description of the threshold (mask) is not

clear. Is it applied to AOD and AE? Why specifically 10% of the maximum? Does that mean that the threshold varies with location, depending on the maximum observed satellite retrieval? Why?

8. Figure S1 shows no satellite observations over land for JFM and limited data for OND, assuming that grey colored areas have zero observations. The picture is worse for AE where no observations are seen over land. With these data points available for evaluation, how can the authors estimate seasonal statistical metrics? This is a clear case where AERONET data and/or additional satellite AOD/AE products should be used to cover for these gaps (MODIS, MISR, or whatever is available for that specific period),

Results

1. Page 9, line 24: The phrase "The seasonal break down is presented besides named." is confusing. Please rephrase.

2. The results section needs some restructuring to allow a nice reading flow. Paragraphs should contain more than one sentence and it will be beneficial to keep the discussion on one topic/figure/statistic in one paragraph. For example, devote one paragraph describing spatial patterns of MODIS AOD, one paragraph for each season and so on.

3. The entire AOD evaluation section does not include an interpretation of the under- or over- estimation seen for each model simulation. Also, why is there a difference between model simulations? This is much more important and interesting than just presenting the statistical findings.

4. Section 3.2, page 18, line 6: "In this section, the simulations run with the available data were less than they were for AOD." What is the meaning of this sentence? Wasn't there one annual simulation performed by each modeling system? This is confusing; please explain.

5. Section 3.2 (Variability): page 22, line 4: why is ES1 showing a remarkable AOD representation? The PDF for ES1 shows that the simulation underrepresented low AOD and overestimated high AOD for the majority of the seasons (except JAS maybe).

6. Figures S5 to S9 in the supplement are never mentioned in the text. In addition, Figure S9 (annual PM2.5 emissions in Europe and North America) does not seem to align with the contents of this manuscript. Please specify the role of all additional figures in the supplementary material.

Summary and Conclusions

1. Page 22, line 29: What is the meaning of front observations in the following phrase "an evaluation of the simulations of the front observations was needed"? Please revise accordingly.

2. Page 23, line 7: Please rephrase the following sentence by replacing "misunderstanding" with a more appropriate characterization ("due to a misunderstanding of the simulation of the aerosol vertical. . .").

3. A lot of the text in section 4 could be included in the respective discussion of the results (see comment #3 in the Results section above). I suggest that the authors discuss here the main key results drawn by their analysis in a clear and concise way.

---

## Author Comment (AC1) · 12 Jun 2018

The authors present the evaluation of the aerosol optical properties simulated in the frame of the Air Quality Model Evaluation International Initiative. As a reference, MODIS AOD is used.
I have several concerns about the paper, which are specified below.
The manuscript needs a major revision before it can be considered for publication in ACP.

**General comments**
The language should be thoroughly checked.
A: The language has been checked by a native proofreader.

The authors say that MODIS "combined" product is not validated (page 8, line 3). If this is true, I would be very careful to evaluate the model results with the product, which is not validated. However, this is not true. Several validation papers for MODIS C6 are published, among them:
http://www.mdpi.com/2072-4292/10/3/475
https://agupubs.onlinelibrary.wiley.com/doi/abs/10.1002/2014JD022453
https://www.atmos-meas-tech.net/6/2989/2013/amt-6-2989-2013.pdf
http://home.iitk.ac.in/~snt/pdf/Mhawish_RSE_2017.pdf
Please, include MODIS validation results and possible bias of the model results comparison, related to bias in MODIS AOD into the discussion.
A: As the referee suggested, the results of the MODIS validation have been included in the manuscript, indicating the information about the error in: section 2.2 Observational Data.

*"The used data from MODIS were Level 2 of the Atmospheric Aerosol Product (both MOD04_L2 and MYD04_L2) from the collection 6, with a resolution of 10 km which are estimated by two different algorithms, Dark Target (DT) and Deep Blue (DB). The used variables were: (1) a "combined" variable of the DT and DB algorithms which provide information about AOD at 550 nm for both ocean and land; and; (2) AE between 550 and 860 nm over the ocean estimated by the DT algorithm. There are several evaluations of this "combined" AOD products of MODIS C6 against AERONET sites around of the world (Sayer et al., 2014; Mhawish et al., 2017; Bilal et al., 2018). All of these established that a high percentage of retrievals are within the estimated error (EE) of the DT and DB algorithms, which is (±0.05 + 15%) (Levy et al., 2013). Moreover, in Sayer et al. (2014) and Bilal et al. (2018) the performance of combined retrievals outperformed DT or DB retrievals in terms of correlation (around 10%), meanwhile relative mean bias values were similar at a global scale. The preliminary estimated error (EE) for the used AE products was 0.45 in the pixels with an AOD > 0.2 (Levy et al., 2013)."*

Moreover, the comparison of model results has been revised in order to include in the discussion the possible model's bias related to MODIS' bias.

MODIS provides very good but not perfect product, and validation results show weaknesses of the AOD product over certain areas. Thus, I strongly recommend to include AOD validation with AERONET (https://aeronet.gsfc.nasa.gov/ ) and MAN (https://aeronet.gsfc.nasa.gov/new_web/maritime_aerosol_network.html ). Otherwise, the results of your analysis are biased by the MODIS quality.
A: As the referee #2 and the referees for the first revision of this work suggested, we have included the AOD and AE validation with AERONET and MAN. The description of the used product has been included in the Observational Data section. Moreover, the Results sections has been re-written taking into account these observations and their results.

Page 8, Line 21-26: *"From AERONET, AOD at 675 nm and AE between 440 and 870 nm retrievals of level 2.0 from the available European stations during the entire 2010 year were used. For this network data, the total uncertainty for the AOD data under cloud-free conditions is established as < ± 0.01 for λ > 440 nm and < ± 0.02 for shorter wavelengths (Holben et al., 1998). The same variables were used from MAN, whose estimates uncertainty of AOD in each channel is, as well as AERONET, < ± 0.02 because MAN is affiliated with the AERONET calibration and data processing standards and procedures (Smirnov et al., 2009)."*

MODIS coverage is surprisingly low in your analysis. How many (in %) pixels you discarded with your mask? Please, show it on the map. Please, repeat the analysis for all reported MODIS AOD pixels. As far as I know, MODIS deliver AOD data which passed the quality control.
A: The number of occurrences in the AOD (Figure S1) and AE (Figure S2) retrievals are showed in the Supplementary Material. We decided to apply "our mask" in order to show temporal means calculated by a number of observations large enough to ensure confident and representative results.

MODIS abroad instruments (Terra and Aqua) in Sun-synchronous orbits around the Earth does not provide data over the entire studied domain for each time step. This, together with the fact that we are using the highest quality of MODIS data, sometimes leads to a small number of occurrences in each pixel. As our results are seasonal means, we cannot calculate the seasonal mean with a really small number of occurrences. So, our mask is established to avoid this fact and in order to build a seasonal mean with a number of observations large enough to make our results robust.

To build our mask we first estimated the maximum of occurrences in the AOD or AE retrievals over the entire domain and period (JFM, AMJ, JAS and OND), which is the maximum number of observations that we can have in our domain. Then, we established the criteria that the pixels shown as final results have to present at least a number of occurrences higher than the 10% of this maximum. So, we show our validation results over the pixels which accomplish with this criterion, but we never discarded any observations over these pixels.

It would not be acceptable to show mean results where means are calculated from a really small number of occurrences, which could lead to a misunderstanding of the validation.

For the new AERONET validation, we followed the same criterion. As AERONET stations are located in a fixed place and we are using hourly means for our evaluation, we establish the maximum of possible occurrences as the maximum of hours in which AERONET stations can retrieve measurements (light hours) during the target periods (JFM, AMJ, JAS and OND). Then, we did not include those stations in which the number of hourly observations is lower than the 10% of the maximum occurrences.

This criterion has not been implemented for the MAN evaluation because this database displays punctual observation with a specific coordinate and time.

We have been re-written Page 8, Lines 27 to 29 in order to improve the explanation of "our mask".

*"All the observations used in this work are not provided in a temporal regularly way. This means that the number of occurrences in each of the pixel for satellite data or in each station for AERONET data are not the same. As the results in this work are represented as seasonal means and in order to show robust means estimated with a reasonable number of occurrences, a mask showing only those pixels (stations) where the satellite (station) occurrences were higher that the 10% of the maximum possible occurrences was implemented. The maximum possible occurrences for satellite data were selected as the maximum of occurrences over the studied season (JFM, AMJ, JAS or OND) and the domain. Figures S1 and S2, in the appendix, show the number of observations used when the mask was implemented. For AERONET, the maximum possible occurrences were established as the maximum of solar-light hours in each station during each season."*

**Specific comments**
Abstract.
- Please, add numbers (abs, or %) for under/overestimation results Done
- I disagree with the conclusion that "spatial and temporal variability of this variable is well-represented by all the models". Is that conclusion done from the comparison with MODIS? As I see from figure S3, almost all models have similar tendencies to underestimate, compared to MODIS, AOD over Siberia and underestimate AOD it the southern part of AOI.

A: We meant that special and temporal variability of AOD was better represented than AE. We have been re-written this sentence:
*"Despite this behaviour, the spatial and temporal variability of AOD was better represented by all the models than AE variability, which was strongly underestimated in all the simulations."*

Page 4, Line 20. Please, correct to "ATSR".Done
Page 7, Line 14. Please, specify "x" in MxD. Done
Page 8, Line 22. What is EE for AE MODIS product? Please, include the discussion to explain your choice for AE between -0.5 and 4.
A: The EE for AE MODIS product is indicated in lines 5-6, page 8: "*The preliminary estimated error (EE) for the used AE products was 0.45 in the pixels with an AOD > 0.2 (Levy et al., 2013).*"
The following discussion about the choice for AE between -0.5 and 4 has been included as:
*"Moreover, and according to the EE for the AE products of MODIS, we set the AE values range between -0.5 and 4.0. It is widely known that AE values spread from 0 to 4 and even sometimes, when really coarse particles are presented, they can reach negative values. Then, we choose AE values between -0.5 as lower limit in order to cover possible negative values in a close smoothing value to the EE for the AE products of MODIS".*

Page 8, lines 4-5. Please, discuss briefly the results here.
A: "The MAE was calculated as in Equation 4 and provides an estimation of the magnitude of the error independently of over-or underestimation."

Page 8, Line 27. Please, explain the mask in the other words. The current explanation is not clear. Was that mask applied for AOD? Or AE? Do you mean, that if for certain location max AOD was 1, you discarded all the cases when AOD<0.1 for that location? Why? For AE, AOD limit of 0.1 is acceptable.
A: As we explained above, we have been re-written Page 8, Lines 27 to 29 in order to improve the explanation of the mask.

Page 8, Line 29. What is the measure of confidence here?
A: The mask is not a proper measure of the confidence of our results. The mask ensures that we are calculating the mean results with a large enough number of occurrences.

**Technical comments**
Please change the color scale for MODIS AOD to max. 1. With the current color scale, the AOD variation below 0.5 is hardly visible. I advice to include also the red color to the color panel.
A: We followed the advice of the reviewer and the maximum of the color scale has been changed. Moreover, we reversed the color scale in order to improve the representation of the AOD variation. We did not include the red color to this panel because this panel contain the green color and people with deuteranopia, protanopia, tritanopia of similar diseases could misunderstanding the color panel. Some examples in this link: https://cran.r-project.org/web/packages/viridis/vignettes/intro-to-viridis.html

Anonymous Referee #3
**General comments**
This is an interesting work that attempts to evaluate modeled AOD and AE over Europe using six model simulations performed under the AQMEII3 framework. The AQMEII initiative has provided a great opportunity for air quality model evaluation and model intercomparison across two continents that allows the community to assess the accuracy of the modeling systems, the drivers of their differences and make suggestions for future model improvements.

The main objection regarding the scientific methodology is the use of only one MODIS product to conduct model evaluation and intercomparison. The authors should take advantage of all possible AOD/AE observations in the model domain over Europe, to enhance their understanding of model behavior.
A: As the referee #3 and the referees for the first revision of this work suggested, we have included the AOD and AE validation with AERONET and MAN.

The paper needs grammatical editing to improve the language and some restructuring to improve the flow. There are a lot of clarifications needed for the methodology and discussion of the results. I am in favor of publishing this paper with Atmospheric Chemistry and Physics with Major Revisions. The specific comments that follow will help improve the discussion of the methodology and significance of the findings so that the overall quality of the manuscript is enhanced.
A: We have revised the entire text and tried to do our best to improve the language and the structure of the text. As above commented, the language had been checked by a native proofreader.

**Specific comments**
Abstract:
   1. Line 13: "this variable" refers to AOD? Please be specific Done

2. General comment: It would be more beneficial to have a quantitative description of the conclusions. If AE is predicted more erroneously than AOD, add some quantitative measures to that statement in terms of errors/biases etc. Descriptive characterizations like "more serious errors" do not add any substantial information.

A: We have corrected this type of expression and added a quantitative description as the referee suggested.

Introduction:
1. Lines 4-5 and elsewhere: please remove the phrase "so-called". The aerosol-cloud- radiation interactions are widely known; the definition of the acronym is enough. Done
2. General comment: The introduction includes a lot of information on past and recent work similar to this study. What is not clear is how this study is different from others. What is the new contribution made by this study to the scientific knowledge of modeled AOD and AE? A high-level brief statement would be appropriate in the introduction.

A: Following the referee suggestion, a brief statement has been included in the last part of the introduction:

*"Curci et al. (2017) used AQMEII Phase 3 simulations to evaluate the black carbon absorption against AERONET but no works have evaluated against satellite the seasonal representation of optical properties over Europe by the regional models involved in AQMEII Phase 3. This represents an added value because three main reasons: 1) all the regional models evaluated here were run using the same boundaries and initial conditions which permit investigate the importance of different processes and feedbacks in each models; 2) the use of two different emissions data set permits evaluated the influence of these in the aerosol optical properties representation; and 3) the use of online coupled chemistry-meteorology/climate models (as were some of the used here) will permit to investigate the influence of the ARI and ACI. As above-mentioned, aerosol optical properties influence ARI and ACI, a good representation of them is, thus, a key issue to reduce the uncertainty of aerosol effects on the Earth's climate system. For this reason, our main study aim was to evaluate the representation of two main aerosol optical properties, AOD and AE, using the models of the AQMEII Phase 3 initiative over Europe. The evaluation was made by using the remote-sensing observations from the MODIS sensor and from AERONET and MAN (Maritime Aerosol Network). Section 2 provides a brief description of the observational and models data, and the evaluation methodology. Section 3 presents the evaluation results. Finally, Section 4 summarises the main conclusions reached."*

Methodology:
1. Page 6, IT1 simulation: ". . .Meteorological inputs were generated using WRF-Chem version 3.4.1. Anthropogenic emission were MACC and biogenic were computed by MEGAN.WRF-Chem was adopted to predict GOCART (Goddard Chemistry Aerosol Radiation and Transport) dust emissions (Ginoux et al., 2001) along with meteorology." What is the role of WRF-Chem in the WRF-CAMx simulations? It is not clear how those two modeling frameworks are combined. It doesn't make sense to use WRF-Chem for meteorology while WRF is already the modeling system used combined with CAMx.

A: There was an error and the referee was right, WRF is used comvined with CAMx and not WRF-Chem. This has been corrected.

2. Table 1 should include one more column that describes the origin of AOD/AE calculations: prognostic (i.e. during runtime) or diagnostic (i.e post-processing).

A: Done

3. Description of dust sources for each simulation is very important since the domain covers North Africa and the signal from the satellite is much stronger there. Prescribed dust emissions (as total PM), online dust sources, etc. This information can be added to Table 1 under Aerosol Mechanism.

A: Done

4. It's a surprise that the authors did not make use of AERONET data for such comprehensive AOD and AE model evaluation. This is strongly recommended for the revised version.

A: As all the referees suggested the evaluation against AERONET and MAN network has been included in the manuscript

5. In addition, the merged AOD product has been used in a number of recent publications with one of the most important being Sayer et al. (2014) (https://agupubs.onlinelibrary.wiley.com/doi/pdf/10.1002/2014JD022453 ) which is a point of reference for the usage of these products.

A: This reference, among others, has been listed as results of the merged AOD product, as well as the error of this product.

6. Observational data: The authors state that "The selection of this observational data was due to results found by Palacios-Peña et al. (2017a)". More detailed explanation is needed for the selection of observational data than the reference to a previous publication. The readers should not have to read other papers to understand the basics of the approach.

A: A more detailed explanation has been done.
*"The selection of this observational data was due to results found by Palacios-Peña et al. (2018) where they evaluated the uncertainty in the satellite representation by comparing MODIS, OMI (Ozone Monitoring Instrument) and SeaWIFS (Sea-viewing Wide Field-of-view Sensor) AOD retrievals against AERONET. They found that MODIS presented the best agreement with the AERONET observations compared to other satellite AOD observations during two studies with high aerosol load took place in 2010 all over Europe."*

7. Observational data (page 8, line 26): The description of the threshold (mask) is not clear. Is it applied to AOD and AE? Why specifically 10% of the maximum? Does that mean that the threshold varies with location, depending on the maximum observed satellite retrieval? Why?

A: The use of the threshold or mask has been explained above. Moreover, we did our best to a better description of the mask in the manuscript.
*"All the observations used in this work are not provided in a temporal regularly way. This means that the number of occurrences in each of the pixel for satellite data or in each station for AERONET data are not the same. As the results in this work are represented as seasonal means and in order to show robust means estimated with a reasonable number of occurrences, a mask showing only those pixels (stations) where the satellite (station) occurrences were higher that the 10% of the maximum possible occurrences was implemented. The maximum possible occurrences for satellite data were selected as the maximum of occurrences over the studied season (JFM, AMJ, JAS or OND) and the domain. Figures S1 and S2, in the appendix, show the number of observations used when the mask was implemented. For AERONET, the maximum possible occurrences were established as the maximum of solar-light hours in each station during each season."*

8. Figure S1 shows no satellite observations over land for JFM and limited data for OND, assuming that grey colored areas have zero observations. The picture is worse for AE where no observations are seen over land. With these data points

available for evaluation, how can the authors estimate seasonal statistical metrics? This is a clear case where AERONET data and/or additional satellite AOD/AE products should be used to cover for these gaps (MODIS, MISR, or whatever is available for that specific period),

A: We have included AERONET in our evaluation.

Results
1. Page 9, line 24: The phrase "The seasonal break down is presented besides named." is confusing. Please rephrase.

A: The phrase has been rephrased as: *"The seasonal means are presented in the named column"*.

2. The results section needs some restructuring to allow a nice reading flow. Paragraphs should contain more than one sentence and it will be beneficial to keep the discussion on one topic/figure/statistic in one paragraph. For example, devote one paragraph describing spatial patterns of MODIS AOD, one paragraph for each season and so on.

A: Following the referee comment, we have restructured the results section.

3. The entire AOD evaluation section does not include an interpretation of the under- or over- estimation seen for each model simulation. Also, why is there a difference between model simulations? This is much more important and interesting than just presenting the statistical findings.

A: Following the referee's comments (here and below), a major effort has been done to take into account the referee's suggestion and the Section 3 (Results) and 4 (Summary and Conclusions) has been restructured.

4. Section 3.2, page 18, line 6: "In this section, the simulations run with the available data were less than they were for AOD." What is the meaning of this sentence? Wasn't there one annual simulation performed by each modeling system? This is confusing; please explain.

A: This phrase has been removed. Instead this has been added at the beginning of the section 3.2: *"AE models simulations are less than for AOD because some models did not provide AOD at different wavelengths and then, it was no possible to estimate AE following the methodology stablished above."*

5. Section 3.2 (Variability): page 22, line 4: why is ES1 showing a remarkable AOD representation? The PDF for ES1 shows that the simulation underrepresented low AOD and overestimated high AOD for the majority of the seasons (except JAS maybe).

A: This ES1 behaviour in the AOD representation is doing in the section Summary and Conclusions.
*"ES1, which used WRF-Chem, presented a high AOD overestimation due to the dust outbreaks. This marked overestimation took place because of a bug in the used dust scheme, which lacks the gravitational settling."*
*…….*
*"For the AOD representation, all the simulations presented similar PDF to the observed values. …. ES1 presented higher probabilities for high AOD values than those observed due to the above-explained lack of dust gravitational settling."*

6. Figures S5 to S9 in the supplement are never mentioned in the text. In addition, Figure S9 (annual PM2.5 emissions in Europe and North America) does not seem to align with the contents of this manuscript. Please specify the role of all additional figures in the supplementary material.

A: For the sake of brevity and in order to show only relevant figures for the work these figures have been remove from the Supplementary Material. Moreover, the figures displayed the total values for the entire year as well as the text regarding they has been removed because they do not provide added valuable information.

Summary and Conclusions

1. Page 22, line 29: What is the meaning of front observations in the following phrase "an evaluation of the simulations of the front observations was needed"? Please revise accordingly.

A: This has been rephrased as: *"an evaluation of the simulations against observations was needed."*

2. Page 23, line 7: Please rephrase the following sentence by replacing "misunderstanding" with a more appropriate characterization ("due to a misunderstanding of the simulation of the aerosol vertical. . .").

A: The phrase has been rewritten: *"As established in Palacios-Peña et al. (2018), this underestimation may be due to a misinterpretation of the simulation of the aerosol vertical and may, therefore, be due to the AOD representation given the understated injection height of the total biomass burning emissions found for the MACC emissions by Soares et al. (2015)."*

3. A lot of the text in section 4 could be included in the respective discussion of the results (see comment #3 in the Results section above). I suggest that the authors discuss here the main key results drawn by their analysis in a clear and concise way.

A: A major effort has been done to take into account the referee's suggestion and the Section 3 (Results) and 4 (Summary and Conclusions) has been restructured.

---

## Author Response (AR3)

Author's Response to the third round of review for the manuscript **"Aerosol optical properties over Europe: an evaluation of the AQMEII Phase 3 simulations against satellite observations (acp-2017-1119)"** submitted by Laura Palacios-Peña

**Report #1: Anonymous Referee #1**

**Suggestions for revision or reasons for rejection**

Q: [A discussion of several underestimation / overestimation issues at certain areas and seasons is going on throughout the paper but there is no explanation related to the atmospheric chemical and physical processes behind these issues…] [The authors should be aware of and use the relevant MODIS products for their comparison. Also, a robust AOD and AE study should consider comparison with AERONET stations which is not present here.]

A: However, we honestly believe there was a little misunderstanding at this stage of review. We were a little bit surprised about Referee #1 comments, because he/she states that there is no comparison with AERONET stations. They were not included in the original submission, but after the first review most of the revised manuscript is devoted to the evaluation of modelling results against MODIS and AERONET observations. The comparison with AERONET was pervasively included in the submitted revised version of the manuscript, as stated in the abstract (page 1, line 10), methodology (page 8, line 29; page 9 and 10), results (page 11: "The numerical result of each case for MODIS (M) and AERONET (A)" and discussion thereafter)". All the Figures include the validation against AERONET stations, and even in the text in the figures we indicate the mean error when compared with AERONET (A). Also, an explanation of the relevant physico-chemical processes is included at the final part of discussion and in the conclusion.

We honestly believe there was all a misunderstanding and the reviewer read the previous (initial) version of the manuscript, which did not include AERONET validation, because all his/her comments are addressed in the revised version we submitted in October.

**Report #2: Anonymous Referee #3**

**Suggestions for revision or reasons for rejection**

Q: The authors have greatly improved the manuscript by following the recommendations from the second revision. The text now reads very well as the language has been improved as well. I have only minor, mostly grammatical, edits that are shown in the list below and I recommend minor revision before publication.

1. Page 1, line 8: replace "remote sense data" with "remote sensing data". Done

2. Page 1, line 16: replace "model election" with "model selection". Done

3. Page 5, lines 13 and 20: please remove the phrase "so-called" in both parts of the text. Done

4. Table 1: the 5th column has as a title the letter "c"; also sometimes "diagnostic" is mispelled as "diasnostic". Please correct. Done

5. Page 8, line 6: "Moreover, in order to conduct a reliable...". Done

6. Page 9, line 12: "rationing" the equation is not an appropriate term. Maybe use "partitioning" instead. Done

7. Page 10, line 10: in the parenthesis "as the ones used in https..." Done

8. Page 10, line 17: correct "spacial". Done

9. Figure 1: explain the difference in the map vs. circles in the figure caption. Corrected caption: "Total and under the mask number of observations used in the analysis. Maps show the number of MODIS observation and point the number of AERONET observations."

10. Page 12, line 8: replace "time means" with "temporal means" or "temporal averages" Done

11. Page 12, line 25: "this affected also the number of..." (remove "to") Done

12. Page 13, line 19: "when versus both MODIS..." does not make sense. Please edit accordingly. Replaced by: "when compared versus both MODIS and AERONET".

13. Page 14, line 27: "seriously the ENSEMBLE..." (remove "in"). Done

14. Page 19, line 30: replace "demanded" with "necessary" or "appropriate". Done

15. Page 21, line 6: please correct "band quiet crudely...". Corrected as "band quite crudely"

16. Figure 2-7: explain in the figure caption the values shown in the insert of every plot (i.e. MAE(M)=0.36, MAE(A)=0,46). In order to clarify, this sentence has been included to captions: "Values in every plot indicate the spatial and temporal average of MBE for MODIS (MBE(M)) and AERONET (MBE(A))."

        A: We really appreciate the kind reviewer's #3 comments. We have revised and corrected point-by-point all his/her edits.